# Projection neurons are necessary for the maintenance of the mouse olfactory circuit

Luis Sánchez-Guardado[1,2]*, Peyman Razavi[1], Bo Wang[1], Antuca Callejas-Marín[1,2], Carlos Lois[1]*

[1]Department of Biology and Biological Engineering, California Institute of Technology, Pasadena, United States; [2]Department of Cell Biology, School of Science, University of Extremadura, Badajoz, Spain

## eLife Assessment

This **valuable** study shows that eliminating a large portion of the principal neurons in the mammalian olfactory bulb does not affect the initial establishment of the circuit but has an impact on its maintenance. The strength of the paper is that the anatomical changes induced by genetic ablation of neurons are clear-cut. There is **solid** support for the findings, with a description of the structural and behavioral effects of ablating the majority of M/T neurons.

*For correspondence:
guardado@unex.es (LS-G);
clois@caltech.edu (CL)

**Competing interest:** The authors declare that no competing interests exist.

**Abstract** The assembly and maintenance of neural circuits is crucial for proper brain function. Although the assembly of brain circuits has been extensively studied, much less is understood about the mechanisms controlling their maintenance as animals mature. In the olfactory system, the axons of olfactory sensory neurons (OSNs) expressing the same odor receptor converge into discrete synaptic structures of the olfactory bulb (OB) called glomeruli, forming a stereotypic odor map. The OB projection neurons, called mitral and tufted cells (M/Ts), have a single dendrite that branches into a single glomerulus, where they make synapses with OSNs. We used a genetic method to progressively eliminate the vast majority of M/T cells in early postnatal mice, and observed that the assembly of the OB bulb circuits proceeded normally. However, as the animals became adults the apical dendrite of remaining M/Ts grew multiple branches that innervated several glomeruli, and OSNs expressing single odor receptors projected their axons into multiple glomeruli, disrupting the olfactory sensory map. Moreover, ablating the M/Ts in adult animals also resulted in similar structural changes in the projections of remaining M/Ts and axons from OSNs. Interestingly, the ability of these mice to detect odors was relatively preserved despite only having 1–5% of projection neurons transmitting odorant information to the brain, and having highly disrupted circuits in the OB. These results indicate that a reduced number of projection neurons does not affect the normal assembly of the olfactory circuit, but induces structural instability of the olfactory circuitry of adult animals.

## Introduction

In order to produce behavior, the brain needs to respond to two contradictory demands. First, to acquire new information, the brain needs to be plastic. Second, to retain information, the brain needs to be stable. The brain of young animals is highly plastic, enabling early experiences to influence the development of many neuronal circuits (*Wiesel and Hubel, 1963*). In contrast, the adult brain is considered to be relatively hard-wired, with limited capacity for morphological changes in response

to natural experiences or different types of experimental manipulations (*Darian-Smith and Gilbert, 1994*; *Florence et al., 1998*).

Two fundamental parameters for the function of brain circuits are the number of neurons that comprise the circuit and how they are connected to each other. Interestingly, within a given circuit and animal species, the number of neurons of a specific cell type in that circuit appears to be relatively conserved. Similarly, there is also significant regularity in the general connectivity trends between neurons in the circuit. These observations have raised the question of whether there is any relationship between the number of a specific type of neuron in the brain, and their patterns of connectivity. In adult *Drosophila*, reducing the number of neurons of specific cell types in the olfactory system does not alter the overall structure of the circuit (*Berdnik et al., 2006*). In contrast, ablation of a particular cell type during embryonic development induces structural changes in the neurites of the remaining neurons of the same type ('peer' cells) in *Drosophila* and mice (*Johnson et al., 2017*; *Wang et al., 2021*). However, less is known about how the reduction in the number of neurons affects the connectivity of their remaining 'peer' cells and how these changes evolve as animals mature, impacting the organization and function of the entire circuit. To further investigate these questions, we focused on the mammalian olfactory bulb (OB).

The structural organization of the olfactory system makes it an ideal model to investigate the mechanisms underlying the connectivity, assembly, maintenance, and plasticity in brain circuits. The olfactory epithelium harbors olfactory sensory neurons (OSNs) that send their axons to the OB. Axons from OSNs expressing the same odor receptor converge into two stereotypic glomeruli within the OB, generating an odor map on its surface (*Mombaerts et al., 1996*; *Ressler et al., 1994*; *Vassar et al., 1994*). The odor information received by the OSNs is transmitted from each glomerulus to cortical areas by the axons of the OB projection neurons, mitral and tufted cells (M/T cells; *Schwob and Price, 1984*; *Walz et al., 2006*). During development, M/T cells initially extend several apical dendrites that innervate multiple neighboring glomeruli. At postnatal stages (P7-P10), M/T cells acquire their mature morphology with the vast majority (>95%) displaying a single apical dendrite innervating a single glomerulus (*Blanchart et al., 2006*; *Hinds, 1968a*; *Hinds, 1968b*; *Shepherd et al., 2004*). Each glomerulus is innervated by the apical dendrites of ~40 peer M/T cells, and it has been demonstrated that peer M/T cells that their apical dendrites project into the same glomerulus differ in their intrinsic excitability and downstream connections (*Burton and Urban, 2014*; *Inokuchi et al., 2017*).

The OB is one of the areas of the adult mammalian brain with the highest levels of structural plasticity. OSNs, glomerular and periglomerular cells are some of the key synaptic partners for M/T cells and they turn over throughout the animal's life (*Graziadei and Graziadei, 1979*; *Merkle et al., 2014*). Surprisingly, despite this continuous turnover, the M/T dendrites are highly stable in adult animals (*Mizrahi and Katz, 2003*). Thus, the synaptic organization of the OB, where each M/T cell, and each set of OSNs expressing the same receptor innervate a single glomerulus, makes the OB circuit an ideal model for understanding how neurons achieve long-term stability.

To investigate the relationship between neuron number and connectivity, we genetically ablated >95% of M/T cells in early postnatal transgenic mice. Despite this drastic reduction in M/T numbers, the initial assembly of the OB circuits proceeded in an apparently normal manner, and the remaining M/T cells refined their arbors to a single dendrite projecting to a single glomerulus, and OSNs expressing the same receptor targeted a single glomerulus. However, as animals matured, the OB wiring became progressively perturbed, and the apical dendrite of remaining M/T cells grew additional branches that innervated multiple glomeruli, indicating that long term stability of M/T cells morphology depends on the presence of peer M/T cells. In addition, after reducing the number of M/T cells the axons of OSNs expressing the same odorant receptor innervated multiple glomeruli, perturbing the olfactory sensory map in the OB surface, indicating that odor map maintenance also depends on a full set of M/T cells. Despite the significant disruptions in the connectivity of the OB circuits, several olfactory-dependent behaviors were preserved, demonstrating the high functional resilience of the olfactory system.

## Results

### Reducing the number of M/T cells soon after birth does not affect the refinement of the apical dendrite of remaining peer cells

M/T cells experience a drastic change in their morphology after birth. During the embryonic and early neonatal period (less than P7), M/T cells have multiple apical dendrites innervating multiple glomeruli. After the first postnatal week, the vast majority (>95%) of M/T cells prune all but one dendrite that innervates a single glomerulus. This refinement can also occur in the absence of sensory input (*Lin et al., 2000*; *Ma et al., 2014*). However, genetic deletion of NMDA receptors in the M/T cells prevents the refinement of their apical dendrites (*Aihara et al., 2021*; *Fujimoto et al., 2023*), indicating that synaptic inputs to M/T cells originating from spontaneous brain activity may regulate the process of dendrite refinement. In the mammalian retina, reducing the number of rod bipolar cells (RBCs) during development triggers dendritic sprouting on remaining RBCs, suggesting that interaction between peer cells sculpts the maturation of their dendritic arbors (*Huckfeldt et al., 2009*; *Johnson et al., 2017*). To investigate whether dendritic refinement of M/T cells depends on their density, we ablated M/T cells soon after birth by crossing the driver *Tbx21*Cre mouse (*Mitsui et al., 2011*) with the reporter *Rosa26*DTA mouse (*Voehringer et al., 2008*; *Figure 1A*). The *Tbx21*Cre mouse expresses cre selectively in M/T cells, and the *Rosa26*DTA mouse expresses diphtheria toxin (DTA) upon cre-mediated recombination, which triggers the death of M/T cells (*Figure 1B*). To count the number of M/T cells we crossed the *Tbx21*Cre mice with the *Rosa26*Ai9 reporter mouse (*Madisen et al., 2010*). The number of (tdTomato (tdT) positive cells) tdT +M/T cells that we observed in the OB of adult Tbx21::Ai9 mice was similar to that previously reported (*Benson et al., 1984*; *Richard et al., 2010*), and by P3 the vast majority, if not all, M/T cells expressed tdT in Tbx21::Ai9 mice at P3 (*Figure 1—figure supplement 1*).

Tbx21::DTA mice survive and reach adulthood without any gross differences from their wild-type siblings (*Figure 1—figure supplement 2A and B* and B). Although the OB volume of Tbx21::DTA mice was reduced to half, the typical OB layer organization was mostly preserved (*Figure 1—figure supplement 2C and D* and D). Interestly, using antibodies against the Tbr2 protein (a marker for M/T cells), we identified a small number of Tbr2+ cells remaining in the mitral cell layer (MCL) and external plexiform layer (EPL) after cell ablation (*Figure 1—figure supplement 2E*). In Tbx21::DTA::Ai9 mice we observed that there was a small percentage of M/T cells that survived and expressed tdT. Previous works have demonstrated that DTA is extremely potent, and that a single molecule of DTA is sufficient to kill a cell (*Yamaizumi et al., 1978*). These observations are consistent with the recombination of the loxP-tdT transgene being more sensitive to cre than the loxP-DTA transgene, a phenomenon that has been routinely reported in the literature (*Becher et al., 2018*; *Kurachi et al., 2019*; *Madisen et al., 2010*; *Song and Palmiter, 2018*). We observed a progressive reduction of M/T cells in Tbx21::DTA::Ai9 mice approximately 25% (P10), 3% (P21), and 1% (P120) of M/T cells remained after cell ablation (and expressed tdT), compared with wild type mice (Tbx21::Ai9 at P120; *Figure 1C*).

Next, we crossed the Tbx21::DTA mouse with the *Rosa26*Confetti reporter mouse to label individual M/T cells with one out four possible fluorescent proteins (*Sánchez-Guardado and Lois, 2019*; *Snippert et al., 2010*). This sparse labeling of cells facilitates investigating whether the refinement of the dendrites of M/T cells depends on interactions between peer cells. At P10, ~95% of the remaining M/T cells had a single apical dendrite innervating a single glomerulus (*Figure 1D*; *Figure 1—figure supplement 3*). These results indicate that the refinement of the apical dendrite of M/T cells proceeds normally despite a progressive drastic reduction in the density of M/T cells.

### M/T cells extend additional dendrites to multiple glomeruli in the absence of neighboring peer cells in adult animals

The decrease in structural plasticity of M/T cells that normally occurs in postnatal mice after the refinement of their apical dendrites (*Mizrahi and Katz, 2003*) is probably due to a combination of cell-autonomous, and non-cell autonomous processes. Cell-autonomous processes depend on genetic program of maturation intrinsic to the individual cells. An alternative model is that the reduction in plasticity with age may be due to non-cell autonomous processes, where each neuron loses plasticity because of the interactions with its neighbors. In normal conditions, despite the turnover of their synaptic partners (OSNs and periglomerular interneurons), the apical dendrite of M/T cells remains very stable (*Mizrahi and Katz, 2003*). To investigate whether the interactions between neighboring

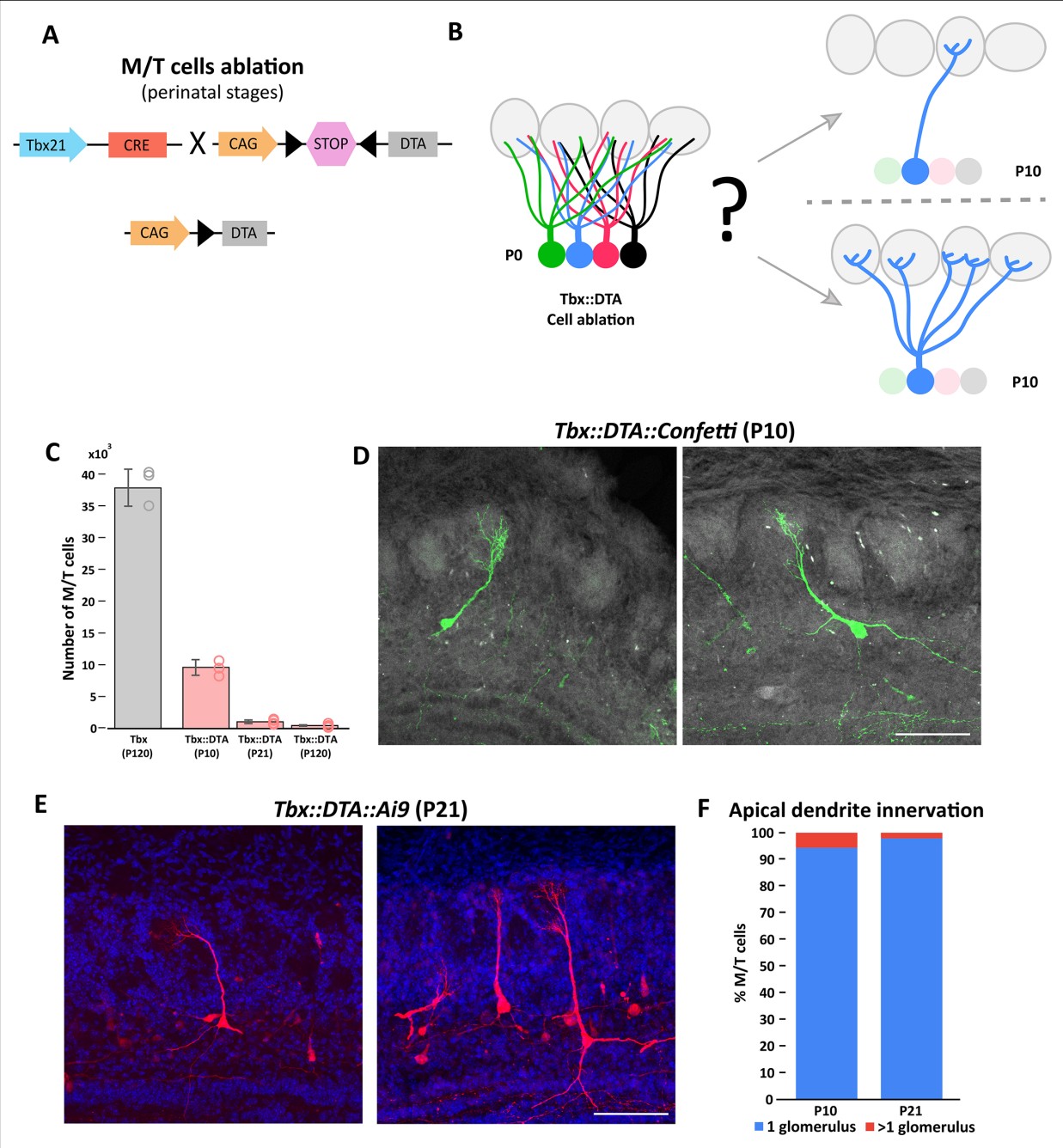

**Figure 1.** The refinement of mitral and tufted cell apical dendrite is not affected by massive cell ablation at early postnatal stages. (**A**) Schematic representation of the transgenic strategy to ablate M/T cells at early postnatal stages. The Tbx21 promoter is specific for M/T cells. The Tbx21 transgene controls the expression of the Cre recombinase in M/T cells, thus regulating the selective expression in M/T cells of the subunit A of the diphtheria toxin (DTA), from a CAG::loxp stop *Rosa26*DTA reporter mouse. (**B**) Schematic representation of the two possible scenarios expected after M/T cell ablation. (top) Reduction of M/T cells density during early postnatal maturation does not affect the apical dendrite refinement. (bottom) The lack of peer M/T cells interferes with the normal refinement of the apical dendrite, thus retaining an immature morphology with multiple apical dendrites. (**C**) Quantification of the number of remaining M/T cells (tdT positive cells) in the OB at different times after early postnatal cell ablation in Tbx21::DTA mice crossed with the *Ai9* reporter mouse. In Tbx21::DTA::Ai9 mice, approximately 25% of the M/T cells remained at P10 (9,606.5±1,229.7; n=4),~3% at P21 (1,024.8±282.3; n=9), and ~1% at P120 (437.6±119.2; n=8) compared with wild type (Tbx21) mice at P120 (38,546.8±2,912.2; n=3). Data are shown as average ± standard deviation. (**D**) Confocal images of individually labeled M/T cells in the OB of a P10 Tbx21::DTA::Confetti mouse after early postnatal cell ablation. Note that M/T cells present a single apical dendrite innervating a glomerulus. Scale bar is 100 µm. (**E**) Confocal images of labeled M/T cells in the OB of a P21 Tbx21::DTA::Ai9 mouse after early postnatal cell ablation. Note that M/T cells present a single apical dendrite innervating a single

*Figure 1 continued on next page*

*Figure 1 continued*

glomerulus. Scale bar is 100 µm. (**F**) Percentage of M/T cells with an apical dendrite innervating one or two glomeruli in P10 and P21 mice after early postnatal cell ablation.

The online version of this article includes the following source data and figure supplement(s) for figure 1:

**Source data 1.** Progressive loss of M/T cells after early postnatal ablation does not affect apical dendrite refinement of remaining M/T cells.

**Figure supplement 1.** M/T cells labeled at early postnatal stages in Tbx21::Ai9 mice.

**Figure supplement 2.** Histological analyses of olfactory bulbs after M/T cells ablation.

**Figure supplement 2—source data 1.** M/T cell ablation in young and adult mice induces changes in animal weight and olfactory bulb volume.

**Figure supplement 3.** M/T cells refine their apical dendrite in the absence of peer M/T cells.

peer M/T cells regulate the stability of their dendrites in adult animals, we reduced the number of M/T cells in full adult animals (2-mo-old) by crossing the Tbx21 mouse with the *Rosa26*<sup>iDTR</sup> mouse (***Buch et al., 2005***). Mice from this cross express the gene for diphtheria toxin receptor (dtR) on the M/T cells. Upon systemic injection of diphtheria toxin (DT), M/T cells are selectively eliminated (***Figure 2A and B***). DT injections into P60 Tbx21::iDTR::Ai9 mice were analyzed two months later (P120), and only 3% of the M/T cells remained compared with the wild type (Tbx21::Ai9) mice (***Figure 2C***).

Next, we analyzed the dendrites of individual M/T cells using a two-component viral vector system to sparse and stochastically label cells that express Cre recombinase in the *Tbx21*<sup>Cre</sup> mice (***Chan et al., 2017***). Viral injections were performed in three types of animals: (i) Tbx21, that had the normal set of M/T cells, (ii) Tbx21::iDTR, in which >90% of M/T cells were killed by DT injection at P60, and (iii) Tbx21::DTA, in which >90% of M/T (cells were killed around P3). In all three conditions, the morphology of M/T cells was examined in adult animals, at P120. As expected, most of M/T cells projected into a single glomerulus in Tbx21 mice (***Figure 2D1***). Surprisingly, in Tbx21::iDTR mice, 48% of the remaining M/T cells branched to several glomeruli (***Figure 2F1***; ***Figure 2—figure supplement 1B***). Interestingly, whereas in Tbx21::DTA mice the apical dendrite of remaining M/T cells innervated a single glomerulus at P10, we also observed that by P120, 36% of remaining M/T cells projected their apical dendrite into multiple glomeruli, as in the Tbx21::iDTR mice (***Figure 2D1***; ***Figure 2—figure supplement 1A***). We also investigated whether the plasticity of the apical dendrite is reduced with age. Cell ablation performed in 6-mo--old Tbx21::iDTR mice (P180) showed that 45% of remaining M/T cells branched to several glomeruli, a percentage comparable to that of animals in which ablation was performed at P60, indicating that the ability of M/T cells to experience drastic morphological changes in their dendrites does not decrease with age (***Figure 2G1***; ***Figure 2—figure supplement 1C***). The tuft length of M/T cells in Tbx21::DTA (early postnatal cell ablation), and Tbx21::iDTR (cell ablation at P60 and P120) was increased compared to apical dendrite tuft in control Tbx21 (***Figure 2—figure supplement 2***). These results indicate that a sufficient density of M/T cells are required for the maintenance of a normal mature morphology (one apical dendrite-one glomerulus), suggesting that structural plasticity is not locked even six months after M/T cells acquired their mature morphology.

Finally, we tested the hypothesis that if glomeruli are innervated by a drastically reduced number of M/T cells this may induce nearby M/T cells to expand their apical dendrites so that they innervate the neighboring empty or depleted glomeruli. In wild type animals, there are ~2000 glomeruli, and each glomerulus is innervated by around 40–50 M/T cells. In animals injected with a high dose of DT (20 µg/kg mouse body weight), as shown above, around 3% (~500) of M/T cells remained, suggesting that in these animals most glomeruli were innervated by very few M/T cells (2000 glomeruli innervated by ~500 M/T cells – roughly 1 M/T for every 4 glomeruli), or none, in many cases. In these animals, around 40% of the remaining M/T cells projected to several glomeruli. When we injected half (10 µg/kg) of the DT dose used in our previous experiments, the fraction of remaining M/T cells was 25% (~10,000 cells) compared to wild type mice (~40,000 cells; ***Figure 2C***). Thus, in animals injected with this lower dose of DT it is expected that each glomerulus would be innervated by around several M/T cells (~10,000 M/T cells for 2000 glomeruli – roughly around 5 M/T per glomerulus). However, in these animals, ~40% of the M/T cells still projected to multiple glomeruli, a similar percentage as in the animals with higher doses of DT in which the vast majority of M/T cells were ablated (***Figure 2H1***; ***Figure 2—figure supplement 1D***). These results suggest that the branching of M/T dendrites into multiple glomeruli does not simply occur because nearby glomeruli are empty. In addition, it is important to emphasize that even in wild type animals with a full set of M/T cells, a small percentage

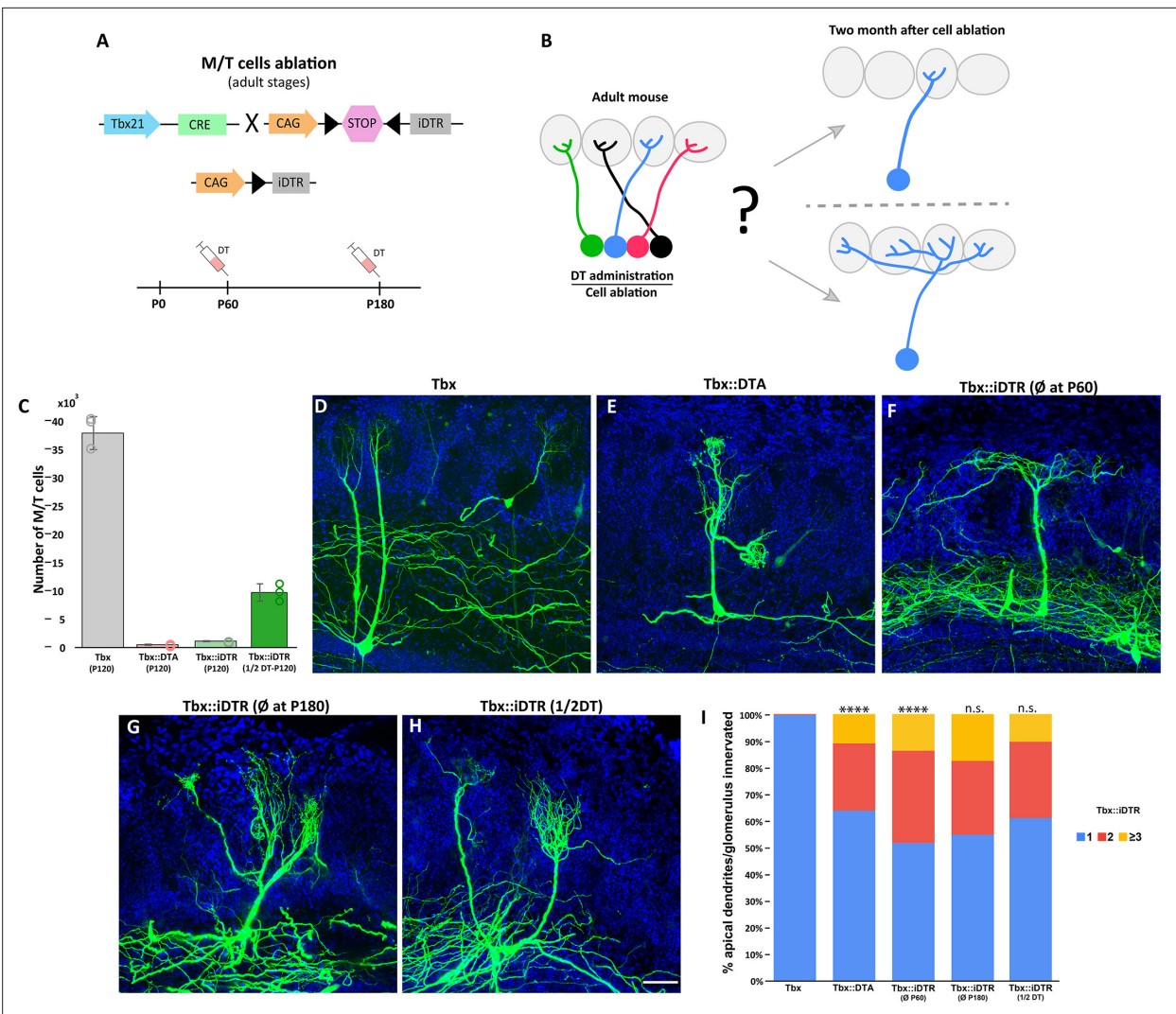

**Figure 2.** Massive ablation of M/T cells in adult mice induces plasticity of apical dendrite. (**A**) Schematic representation of the transgenic strategy to ablate M/T cells in adult animals. The Tbx21 promoter is specific for M/T cells. The Tbx21 transgene controls the expression of the Cre recombinase in M/T cells, enzyme, thus regulating the selective expression in M/T cells of the receptor (iDTR) for diphtheria toxin (DT), from a CAG:loxp stop *Rosa26*[iDTR] reporter mouse. Upon systemic injection of DT into the bloodstream, DT gets transported into M/T cells expressing iDTR, and selectively kills them. (**B**) Two possible scenarios are expected after ablating most of the M/T cells in adult animals. (top) M/T cells density reduction does not induce changes in the apical dendrite structure. (bottom) The absence of peer M/T cells induces structural plasticity in remaining M/T cells, whose apical dendrites extend into nearby glomeruli. (**C**) Quantification of the number of remaining M/T cells (tdTomato [tdT] positive cells) in the OB of adult mice (P120) in different cell ablation experiments. 437.6±119.2 M/T cells remained in Tbx21::DTA::Ai9 mice (~1% of the full set of M/T counted in wt mice; n=8; light red). 1158±99 M/T cells remained in Tbx21::iDTR::Ai9 mice (~3% compared to wt mice; n=4; light green). 9819±1544 M/T cells remained in Tbx21::iDTR::Ai9 mice when half of the DT concentration (10 µg/kg of body weight) was injected (~25% compared to wt; n=3; dark green). Wild type (Tbx21::Ai9) mice had 38,546.8±2912.2; n=3; gray. Data are shown as average ± standard deviation. (**D–H**) Confocal images of M/T cells after sparse labeling using AAV-PHP.eB-DiO-GFP. (**C**) *Tbx21* (wild type) mice at P120. (**D**) Tbx21::DTA mice at P120. (**E**) Tbx21::iDTR mice at P120 (cell ablation induced at P60). (**F**) Tbx21::iDTR mice at P240 (cell ablation induced at P180). (**G**) Tbx21::iDTR mice at P120 (cell ablation induced at P60 using half of the DT dose). Blue = DAPI staining. Scale bar in H is 50 µm and applies to (**D-H**). (**I**) Percentage of M/T cells with an apical dendrite innervating one, two, three or more glomeruli for experimental conditions in D-H. In Tbx21::DTA ~35% of M/T cells had an apical dendrite innervating more than one glomerulus, while in control mice (Tbx21) ~99% M/T cells had an apical dendrite innervating a single glomerulus (Tbx21 (n=158) and Tbx21::DTA (n=149); p-value <0.0001; $\chi 2$ test). In Tbx21::iDTR mice, when ablation was induced at P60, 48% of M/T cells had an apical dendrite innervating more than one glomerulus (n=188; p-value = <0.0001, compared to Tbx21), but no significant differences were observed when compared to Tbx21::DTA (p-value = 0.05). No significant differences were observed in Tbx21::iDTR mice when comparing cell ablation performed at P60 versus P180 (n=236; p-value = 0.15) or P60 versus M/T cells ablation using half-dose of DT at P60 (n=308; p-value = 0.05).

The online version of this article includes the following source data and figure supplement(s) for figure 2:

**Source data 1.** M/T cell density reduction induces changes in the apical dendrite structure.

*Figure 2 continued on next page*

*Figure 2 continued*

**Figure supplement 1.** Plasticity of M/T cells apical dendrite is induced by the reduction of peer cells.

**Figure supplement 2.** Tuft length of remaining M/T cell apical dendrites.

**Figure supplement 2—source data 1.** Apical dendrite tuft length of remaining M/T cells.

of M/T cells still innervate more than one glomerulus (*Lin et al., 2000*). Together, these observations suggest that the innervation of multiple glomeruli by M/T cells is not simply due to the presence of empty glomeruli.

These results indicate that the ability of M/T cells to grow new dendrites, and to project them to multiple glomeruli is not restricted to neonatal stages and that it can be induced in adults (6 mo old) mice, simply by reducing the density of their peer M/T cells. Interestingly, only around half of the surviving M/T cells their apical dendrites extended toward several glomeruli, while the other half maintained a single dendrite that targeted a single glomerulus. This observation suggests that the ability of M/T cells to change their structure and grow dendrites into adjacent glomeruli may be restricted to specific subtypes of M/T types.

## Mechanisms controlling M/T cells structural plasticity in adult animals

Activity plays an important role in sculpting cortical sensory circuits (*Dräger, 1978*; *Gordon and Stryker, 1996*; *Sawtell et al., 2003*). However, M/T cells apical dendrite refinement occurs in the absence of sensory activity (*Lin et al., 2000*; *Ma et al., 2014*), indicating that it is independent of input from OSNs. To investigate whether the structural plasticity induced in remaining M/T cells after ablation is regulated by sensory input, we performed unilateral naris occlusion (*Figure 3A*). Sensory deprivation was carried out in P50 Tbx21::iDTR mice, starting 10 d before M/T cell ablation. We used a two-component viral vector system to sparse and stochastically label cells, which allowed us to visualize the dendrites of individual M/T cells (*Chan et al., 2017*). OBs were analyzed 2 mo after naris occlusión (P120). Sensory deprivation causes a decrease of tyrosine hydroxylase (TH) expression in PG cells, and we confirmed that in these animals sensory deprivation was effective by TH immunostaining (*Figure 3—figure supplement 1*). M/T cells tuft length in sensory deprived OBs was reduced compared to apical dendrite tuft in control Tbx21::iDTR mice (ablation at P60 and unmanipulated sensory input; *Figure 3—figure supplement 4*). However, the percentage of M/T cells innervating multiple glomeruli was similar regardless of whether the animals had a closed (39%) or open naris (44%) compared to the Tbx21::iDTR mice (48%; control mice; *Figure 3B and E*; *Figure 3—figure supplement 2A* and B). This result suggests that sensory stimuli are not essential for the expansion of M/T apical dendrites into multiple glomeruli in remaining M/T cells.

To examine the role that the spontaneous activity of M/T cells may play in their apical dendrite plasticity in the absence of peer M/T cells, we overexpressed the inwardly rectifying potassium channel (Kir2.1) which reduces the firing of action potentials (*Figure 3C*; *Burrone et al., 2002*). Kir2.1 channel was delivered into Tbx21::iDTR mice 2–3 weeks before DT injection at P60 using a viral vector in which Kir expression depended on cre recombination. Thus, in this strategy, Kir was selectively expressed in M/T cells, and as expected, the firing of M/T cells was strongly reduced in the expressing cells Kir2.1 (*Figure 3—figure supplement 3*). As observed with sensory deprivation, Kir2.1 expression caused a significant reduction of the tuft length in the apical dendrite of remaining M/T cells (*Figure 3—figure supplement 4*). However, the overexpression of the Kir2.1 channel in remaining M/T cells did not cause statistically significant differences in the percentage of M/T innervating multiple glomeruli compared to remaining M/T cells in Tbx21::iDTR:DTA mice (*Figure 3D and E*; *Figure 3—figure supplement 2C and D*). These experiments demonstrate that the structural plasticity observed after reducing the total number of M/T cells in an animal is not regulated by the firing action potentials in the remaining cells. Instead, this experiment indicates that the observed structural plasticity may be regulated by other types of mechanisms (such as increased synaptic excitation) that do not require the M/T cells to fire action potentials in a physiological manner.

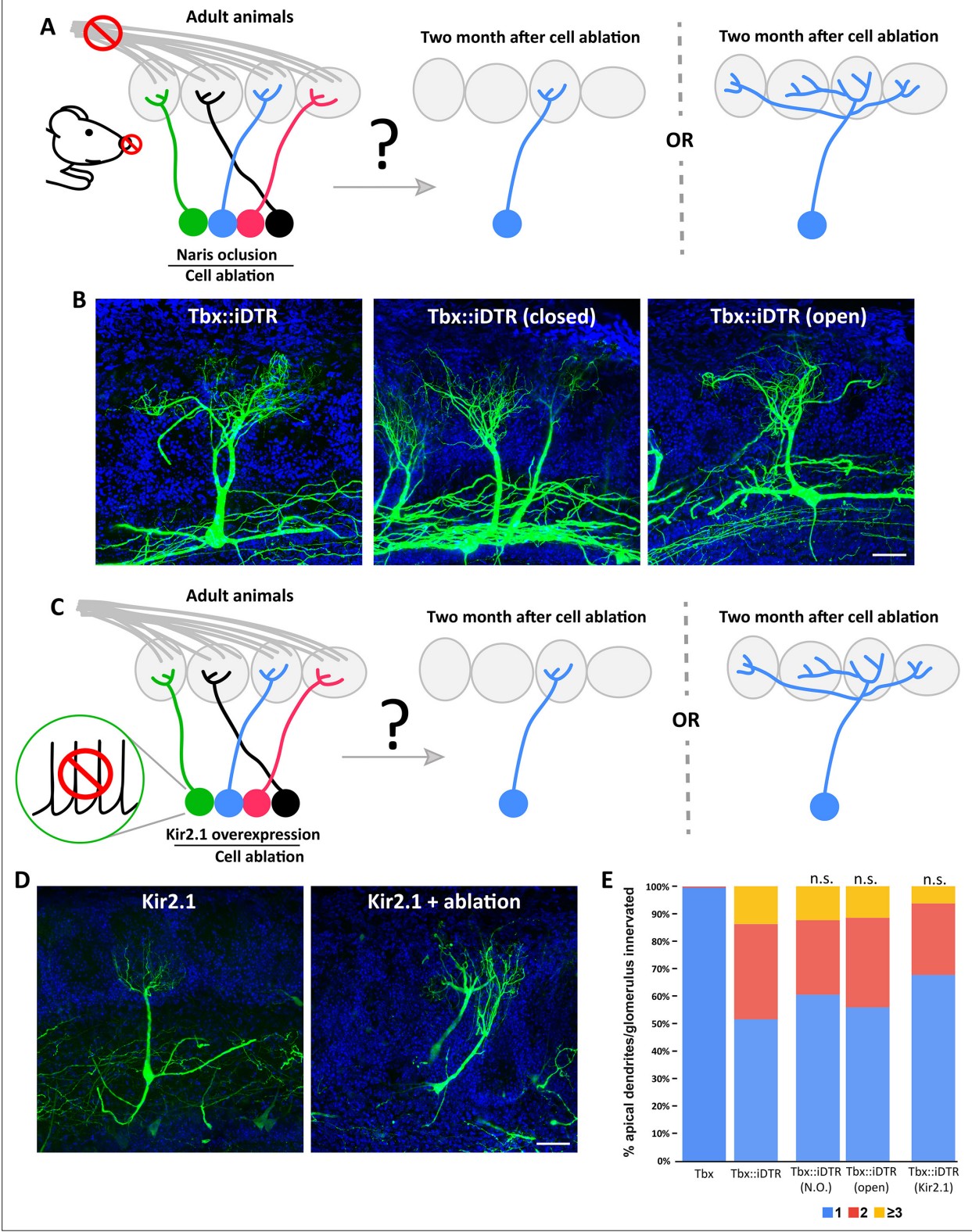

**Figure 3.** M/T cell apical dendrite plasticity caused by reduction in the number of neurons is independent of neuronal activity. (**A**) (left) Schematic representation of the strategy to perform sensory deprivation (naris occlusion) in Tbx21::iDTR prior to cell ablation induced at P60. (right) Two possible scenarios are expected in the remaining M/T cells of sensory deprived OBs. (left cartoon) Sensory deprivation prevents the structural changes of M/T caused by ablation of M/T peers and retain a single apical dendrite, or (right cartoon) M/T cells undergo structural plasticity in sensory deprived OB, as shown in *Figure 2F*. (**B**) Confocal images of M/T cells labeled using AAV-PHP.eB-DiO-GFP showing that remaining M/T cells in the OBs sensory

*Figure 3 continued on next page*

*Figure 3 continued*

deprived by naris closure (middle) in Tbx21::iDTR mice still remodeled their apical dendrites, such that they branch toward multiple nearby glomeruli, as observed in the contralateral OB with an open naris (right), or in control Tbx21::iDTR mice in which both nares were left open (left). Blue = DAPI staining. Scale bar in B is 50 μm. (**C**) Schematic representation of the strategy to selectively reduce neuronal activity in M/T cells in Tbx21::iDTR by expressing the Kir2.1 channel before inducing cell ablation at P60. Two possible scenarios are expected in the remaining M/T cells. Neuronal activity reduction prevents the structural changes of M/T caused by ablation of M/T peers and retain a single apical dendrite, or M/T cells with reduced neuronal activity undergo structural plasticity in sensory deprived OB, as shown in *Figure 2F*. (**D**) Confocal images of M/T cells labeled using AAV-PHP.eB-DiO-Kir2.1–2 A-GFP show that M/T cells expressing the Kir2.1 channel in the remaining cells when cell ablation was performed in Tbx21::iDTR mice, showed structural changes in their apical dendrite (right). M/T cells expressing the Kir2.1 channel without cell ablation maintained their normal morphology, with a single apical dendrite innervating a single glomerulus. Blue = DAPI staining. Scale bar in D is 50 μm. (**E**) Percentage of M/T cells with apical dendrites innervating one, two, three or more glomeruli for experimental conditions shown B and D. Similar results were obtained in M/T cells from both OBs, closed (38%; n=231) or open (44%; n=228) nostril. No significant differences were observed between Tbx21::iDTR mice with open versus closed nostrils (p-value = 0.17; $\chi$ 2 test). Similarly, in Tbx21::iDTR animals that received a DT injection (Tbx21::iDTR +DT) at P60 the expression of the Kir2.1 channel (Tbx21::iDTR +DT + Kir) did not cause any significant changes in the morphology of M/T cells (p-value = 0.05).

The online version of this article includes the following source data and figure supplement(s) for figure 3:

**Source data 1.** Neuronal activity does not regulate the plasticity of apical dendrites.

**Figure supplement 1.** Confirmation of sensory deprivation by naris occlusion.

**Figure supplement 2.** M/T cell apical dendrite plasticity caused by reduction in the number of neurons is independent of neuronal activity.

**Figure supplement 3.** M/T cells activity reduction by overexpressing the Kir2.1 channel.

**Figure supplement 3—source data 1.** Overexpressing the Kir2.1 channel reduces M/T cell activity.

**Figure supplement 4.** Apical dendrite tuft length of remaining M/T cell with reduced activity.

**Figure supplement 4—source data 1.** Apical dendrite tuft length of remaining M/T cells with reduced neuronal activity.

## Odor map disruption caused by a drastic reduction in the number of M/T cells

OSNs send their axons from the olfactory epithelium to the OB. During development, axons from OSNs expressing the same odor receptor initially project to several neighboring glomeruli. In the first postnatal week, OSNs axons undergo a refinement process such that all OSNs expressing one receptor project into a single glomerulus and form a stereotypic odor map on the OB surface (*Mombaerts et al., 1996*; *Ressler et al., 1994*; *Vassar et al., 1994*; *Wang et al., 1998*).Once the odor map is established, it is maintained throughout the animal's lifespan, despite the constant turnover of OSNs (*Zou et al., 2004*). Above, we described that reducing the density of M/T cells in Tbx21::DTA and Tbx21::iDTR mice breaks the 'one apical dendrite-one glomerulus' rule, such that remaining M/T cells branched into several glomeruli. Thus, we investigated whether M/T cells could also play a role in the refinement and maintenance of the odor map from the OSNs into the bulb.

To study whether M/T cells could regulate OSN refinement, we crossed the Tbx21::DTA mouse (to produce early postnatal cell ablation) with the *Or8a1b*ChR2-YFP mouse (M72 for short; *Smear et al., 2013*); which expresses yellow fluorescent protein under the promoter of the *Or8a1b* odor receptor (M72 odorant receptor; *Figure 4A and B*). At P10 and P21 there were no differences in the targeting of M72 axons between Tbx21::M72 and Tbx21::DTA::M72 mice (*Figure 4C and E*). These results suggest that a normal density of M/T cells is not necessary to refine OSN axons and odor map formation, and it is consistent with previous works (*Bulfone et al., 1998*).

Next, we investigated whether M/T cells are necessary for odor map maintenance by ablating M/T cells in adult Tbx21::iDTR::M72 mice (ablation induced by DT injection at P60). Surprisingly, M72 axons projected to several glomeruli when OBs were analyzed 2 mo later (P120; *Figure 4D and E*). Interestingly, whereas in Tbx21::DTA mice M72 axons projected into a single glomerulus at P10 and P21, we also observed that by P120 the M72 axons projected into multiple glomeruli, as in the Tbx21::iDTR mice (*Figure 4D and E*). These results suggest that the maintenance of the odor map on the OB surface depends on the presence of a sufficient density of M/T cells.

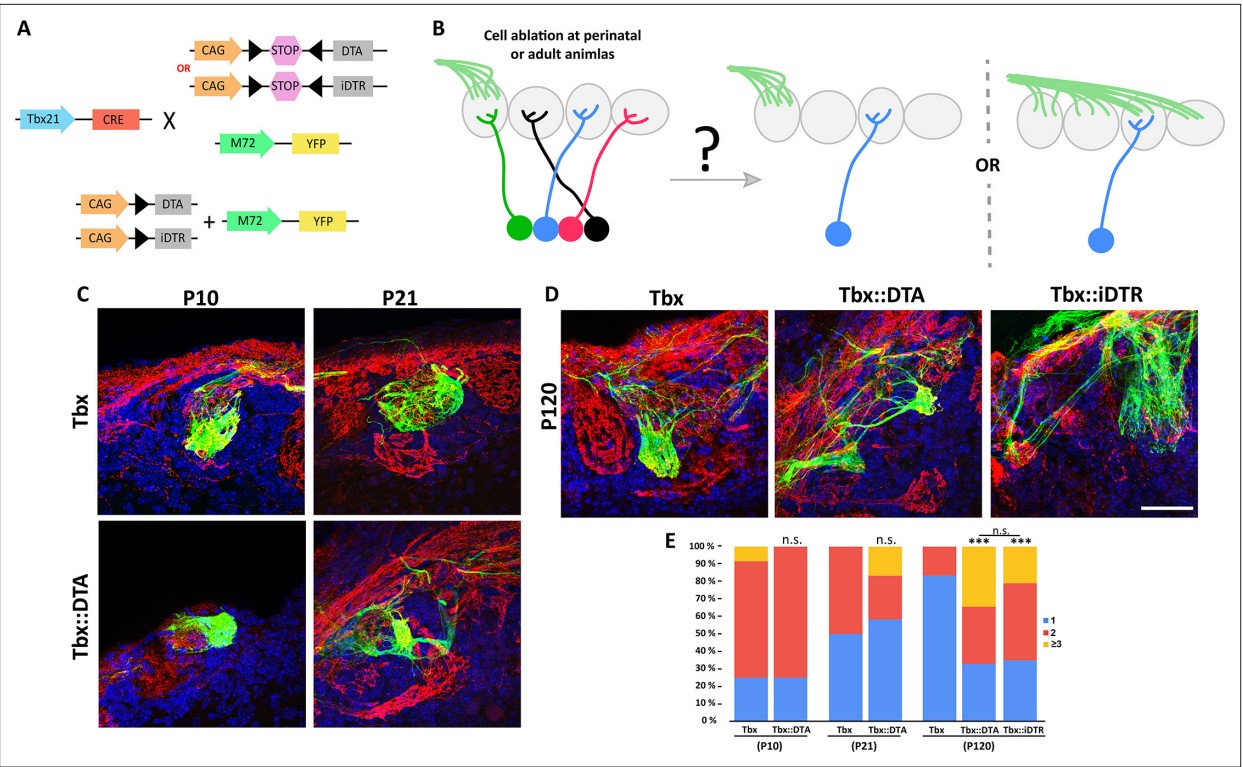

**Figure 4.** Odor map maintenance is disrupted by the drastic reduction in the number of M/T cells. (**A**) Schematic representation of the transgenic strategy to perform cell ablation in early postnatal (Tbx21::DTA) and adult (Tbx21::iDTR) mice to analyze the axon projection of the M72 olfactory sensory neurons into the OB. (**B**) Two possible scenarios are expected for the OSN axon projection to the OB after M/T cells ablation. (left) M72 OSN axon project to a single glomerulus per hemibulb as in wt animals, or (right) M72 OSN axons project to more than one glomeruli per hemibulb, perturbing the odor map in the OB surface. (**C**) Confocal images of M72 OSN axons (green) in a wild type mouse (Tbx21, top row) and Tbx21::DTA mice (bottom row). Each column represents the analysis of the OSN axon at two different time points (**P10 and P21**). Immunostaining against olfactory marker protein (OMP, red) labels all OSN axons. Blue = DAPI staining. (**E**) Confocal images of M72 OSN axons (green) in Tbx21, Tbx21::DTA and Tbx21::iDTR mice at P120. The glomerular projection was disrupted in adult mice, both in animals in which ablation occurred soon after birth (Tbx21::DTA) or as adults (Tbx21::iDTR injected at P60 with DT). Immunostaining against OMP (red) labels all OSN axons. Blue = DAPI staining. Scale bar in D is 50 μm and applies to (**A-D**). (**F**) Percentage of M72 OSN axons projecting to one, two, three or more glomeruli per hemibulb in Tbx21, Tbx21::DTA and Tbx21::iDTR mice. No differences were observed at P10 and P21 between the M72 OSN axon in normal conditions or with reduced M/T cells Tbx21: P10 (n=12), P21 (n=20); Tbx21::iDTR: P10 (n=12), P21 (n=12). At P10 (p-value = 0.59; $\chi$2 test), P21 (p-value = 0.23). However, significant differences in M72 OSN axons were observed in Tbx21::DTA (n=64; p-value <0.001) and Tbx21::iDTR (n=52; p-value <0.001) mice at P120, projecting to multiple glomeruli instead of a single glomerulus as in Tbx21 mouse (n=24).

The online version of this article includes the following source data for figure 4:

**Source data 1.** Disruption of the odor map after M/T cell ablation.

## Olfactory resilience after a drastic reduction in the density of bulb projection neurons

### Learned olfactory behaviors: odor detection and discrimination

In rodents, many behaviors critically depend on olfaction. Previous experiments with mutant mice affecting the olfactory system have reported severe perturbations of behavior, including the inability to find food or choose mates (*Belluscio et al., 1998*). Tbx21::DTA and Tbx21::iDTR mice not only have very few M/T cells remaining (fewer than 5%), but also the architecture of their OBs is disrupted: OSNs expressing the same odor receptor project into several glomeruli, and the apical dendrite of individual M/T cells innervate multiple glomeruli. Thus, we investigated the ability of *Tbx21::DTA* (early postnatal cell ablation) and *Tbx21::iDTR* (adult cell ablation) mice to identify odors using a go/no-go paradigm (*Bodyak and Slotnick, 1999*). Despite the massive loss of M/T cells and the disorganization of OB connectivity, all Tbx21::DTA mice reached 80% accuracy when exposed to 5% cineole (a chemical typically used for olfactory tests) concentrations on their first trial and even showed comparable accuracy to control Tbx21 mice at 0.5% cineole concentrations (*Figure 5A*). Whereas all the Tbx21::DTA

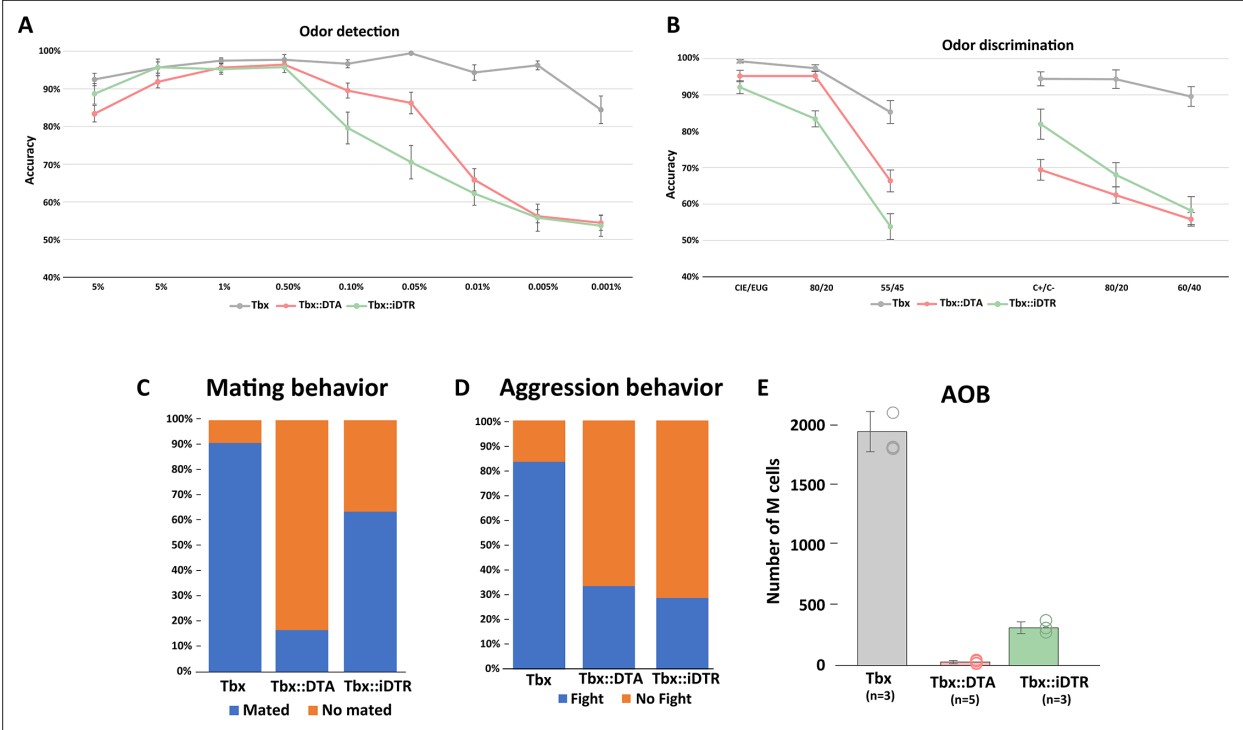

**Figure 5.** Mouse behavior after massive M/T cells ablation. (**A**) Olfactory performance of Tbx21 (n=12), Tbx21::DTA (n=13), and Tbx21::iDTR (n=11) mice in a go/no go paradigm to investigate their odor detection ability. Decreasing cineole concentrations were presented each experimental day. Dots represent the mean percentage of accuracy responses in a single day. Accuracy lower than 80% was considered as a deficit in olfactory detection. Tbx21::DTA and Tbx21::iDTR mice successfully performed odor detection with concentrations of cineole as low as 0.05%, but failed when the concentration of odorant was further reduced. Control mice (Tbx21) successfully detected cineole at all concentrations tested (up to 0.001%). (**B**) Odor discrimination experiment exposing mice to increasing binary mixtures of cineole and eugenol (left) and (+)-carvone and (−)-carvone odors (right). Both experimental mice (Tbx21::DTA and Tbx21::iDTR) could successfully detect cineole and eugenol when they were presented as individual odors, or with 80/20 mixtures. However, they failed when mixtures of cineole::eugenol were 55/45. However, Tbx21::iDTR mice discriminated between (+)-carvone and (−)-carvone odors presented individually, but not when odor mixtures were presented. Tbx21::DTA could not discriminate between (+)-carvone and (−)-carvone even when they were presented as individual odors. (**C**) Tbx21::DTA and Tbx21::iDTR males mated with wt females with a much lower frequency (Tbx21::DTA = 16.67%; n=12; and Tbx21::iDTR = 63.63%; n=11) compared to control males (90.91%; n=11). (**D**) In a resident/intruder paradigm, then the resident and intruder were both wt male mice, the resident started a fight in >90% of the tests. When the resident was a wild type male they fought in 33.33% of the cases if the intruder was a Tbx21::DTA male (n=9) or 28,57% if the intruder was a Tbx21::iDTR male (n=7). When the residents were Tbx21::DTA (early postnatal cell ablation) or Tbx21::iDTR (cell ablation at P60) males, they did not fight when the intruder was a male with the same genotype or a wild type male mouse. (**E**) Quantification of M/T cells in the accessory olfactory bulb (AOB) in Tbx21, Tbx21::DTA, and Tbx21::iDTR showed that only 1.6% of M/T cells remained (29.4±13.7, average ± S.E.M.; n=5) in Tbx21::DTA mice and 16,6% in Tbx21::iDTR (315.3±50; n=3) compared to control Tbx21 mice (1900±168.4; n=3).

The online version of this article includes the following source data and figure supplement(s) for figure 5:

**Source data 1.** Reduction of M/T cell numbers affects mouse behavior.

**Figure supplement 1.** Odor detection accuracy after reduction of M/T cells in adult mice.

**Figure supplement 1—source data 1.** Odor detection accuracy after M/T cells depletion in adult mice.

mice (with early postnatal M/T ablation) succeeded in the go/no-go test, only 50% of Tbx21::iDTR (in which M/T ablation occurred as adults) successfully performed this paradigm on the first day of testing. Of the remaining 50% of Tbx21::iDTR animals, 21% of the mice were completely unable to perform this detection test at any time, even after training on this test for 3 weeks. The remaining 29% of Tbx21::iDTR mice were eventually able to complete the test after 1 week of training (*Figure 5A*; *Figure 5—figure supplement 1*). These observations indicate that the assembly of an olfactory circuit with a reduced density of projection neurons in young animals (Tbx21::DTA mice) is compatible with preserved olfactory performance. In contrast, reducing the density of M/T cells in an adult animal after the circuitry of its olfactory bulb was assembled normally (in Tbx21::iDTR mice) causes important deficits in olfactory performance; however, in some cases, these deficits can be overcome after training.

After the animals were exposed to the detection test, we investigated the ability of these mice to perform a more challenging olfactory test: to discriminate between two odors with very different chemical properties (cineole (S+) and eugenol (S-)). Both Tbx21:DTA and Tbx21::iDTR mice reached more than 80% accuracy criterion when exposed to pure molecules and binary mixtures (80/20::20/80; *Figure 5B*). However, they had difficulties discriminating between more similar binary mixtures (55/45::45/55). Next, Tbx21::DTA and Tbx21::iDTR mice were exposed to highly similar smells, such as the monomolecular enantiomers (carvone (+) (S+) and carvone (-) (S-)). Tbx21::DTA mice failed to discriminate between carvone (+) and carvone (-), even when the odorants were presented as pure molecules (not as mixtures). In contrast, Tbx21::iDTR mice reached 80% accuracy in this version of the discrimination test with pure molecules (*Figure 5B*). We should note that both in Tbx21::DTA and Tbx21::iDTR animals, the discrimination test (using cineole (S+) and eugenol (S-)) was performed after they were exposed to the detection test (using cineole (S+) and mineral oil (S-)). Thus, it is possible that the good performance of some the Tbx21::iDTR animals in the discrimination test could be due to the prior training in the detection test with the cineole as the reworded odor (S+), in agreement with the improved performance of the Tbx21::iDTR animals on the detection test after several days of practice, as indicated above.

These results suggest that despite the reduced number of M/T cells and disorganization of their OBs in Tbx21::DTA and Tbx21::iDTR mice, their olfactory system has a remarkable resilience that partially preserves their ability to smell. However, it is important to note that these standard tests commonly used to measure olfactory function require substantial training, and thus, are quite artificial.

## Innate olfactory-driven behaviors: mating and aggression

In rodents, many key social behaviors depend on olfaction. In mice, litters are primarily taken care of by females. No apparent differences were observed in offspring nursing between both experimental Tbx21::DTA and Tbx21::iDTR mice, indicating that this behavior was mostly preserved in females. However, the reduction of M/T cells affected the mating rate of Tbx21::DTA and Tbx21::iDTR male mice. Individual Tbx21::DTA and Tbx21::iDTR male mice were housed with a single wild type female for three nights. We scored that a male mated with a female when a vaginal plug was observed. Only 16.67% of Tbx21::DTA males mated, compared to 63.63% of Tbx21::iDTR and 90.91% control Tbx21 animals (*Figure 5C*). Next, we used a resident/intruder paradigm to analyze the aggressive behavior of the male mice. Tbx21::DTA and Tbx21::iDTR mice did not initiate a fight when a wild type male or a male mouse from the same strain (Tbx::DTA or Tbx21::iDTR)was introduced into its cage. Moreover, when Tbx21::DTA or Tbx21::iDTR male mice were the intruders, the resident wild type mice attacked them in 33% (Tbx21::DTA) and 28% (Tbx21::iDTR) of the cases, compared to 83% when the intruder was another wild type male mouse (*Figure 5D*).

The accessory olfactory bulb (AOB) plays a key role in innate social olfactory behaviors. We quantified the number of M/T cells remaining in the male AOB. Wild type mice have around ~2000 M/T cells in their AOBs, while only ~1.5% and 16% of M/T cells remained in the AOB of Tbx21::DTA::Ai9 mice and Tbx21::iDTR::Ai9 mice, respectively (*Figure 5E*). Thus, the drastically reduced number of AOB M/T cells in the Tbx21::DTA and Tbx21::iDTR mice could explain their inability to engage in social behaviors such as a mating or aggression, suggesting that behaviors depending on the AOB are hard-wired, less plastic, and less resilient than the learned behaviors that depend on the main OB.

# Discussion
## A full set of M/T cells is necessary for the maintenance of a stable dendrite

Dendrite morphology is one of the main morphological features of a neuron. The shape and size of the dendritic arbor are important determinants of the number and types of inputs that a neuron receives (*London and Häusser, 2005*). During development, each M/T cell extends several apical dendrites toward several glomeruli at the same time that OSN axons penetrate into the OB surface to form a glomerular structure (*Blanchart et al., 2006*). In the first week after birth, M/T cells undergo a refinement process where they end up with a single apical dendrite innervating a single glomerulus (*Blanchart et al., 2006*; *Hinds, 1968a*; *Shepherd et al., 2004*). It is known that OSN axons are required both for glomerular formation and for the refinement of the apical dendrite of M/T cells

(*Kobayakawa et al., 2007*; *Nishizumi et al., 2019*). In the absence of OSN axons, M/T cells maintain an immature morphology with multiple apical dendrites even as animals mature and become adults (*Kobayakawa et al., 2007*; *Nishizumi et al., 2019*). Despite the requirement of OSN axons to form a glomerular structure, it has been described that the refinement of the apical dendrite of M/T cells is independent of the sensory experience information driven by OSNs inputs. M/T cells apical dendrite refinement is delayed by a few days when OSNs neuronal activity is reduced or neurotransmitter release blocked (*Aihara et al., 2021*; *Fujimoto et al., 2023*; *Lin et al., 2000*; *Lorenzon et al., 2015*; *Ma et al., 2014*; *Matsutani and Yamamoto, 2000*; *Nishizumi et al., 2019*).

Recently, studies have shown that the refinement of the apical dendrites of M/T cells is prevented when the glutamatergic NMDA receptors are deleted (by knocking out the *Grin1* gene) or by silencing the electrical activity of M/T cells (by electroporating Kir2.1; *Aihara et al., 2021*; *Fujimoto et al., 2023*). The glutamatergic input to M/T cells could originate from multiple sources, including external tufted cells, axonal inputs from the olfactory cortex, or from reciprocal connections between M/T cells. In addition, the refinement of the apical dendrite of M/T cells was also reduced when the synaptic vesicle release from M/T cells was blocked by expression of tetanus toxin (*Fujimoto et al., 2023*). This observation suggests that release of glutamate between M/T cells at early postnatal stages might regulate the refinement of their apical dendrites (*Fujimoto et al., 2023*).Based on tdT expression in Tbx21::Ai9 mice, expression of cre started around P0-3 (*Figure 1—figure supplement 1*). Accordingly, in Tbx21::DTA mice one might expect that M/T cells would start to die also around this time. However, our results indicate that the refinement of the apical dendrite of M/T cells proceeds normally, although the number of M/T cells is progressively eliminated soon after birth. Thus, it is possible that we did not observe any obvious effect in the process of dendrite refinement because in Tbx21::DTA mice the apical dendrites had completed their refinement around P10, a time when 25% of M/T cells are still alive in Tbx21::DTA mice.

Once the M/T cell apical dendrite refinement process is complete such that each M/T cell projects a single apical dendrite to a single glomerulus, this status stays stable throughout the life of the animal (*Ma et al., 2014*; *Mizrahi and Katz, 2003*). In the OB, each glomerulus is innervated by approximately 40 M/T peer cells where they make synapses with OSN axons expressing the same odor receptor, juxtaglomerular cells (external tufted cells [ETCs] and periglomerular cells [PGCs], and short axon cells [SACs]). Despite the continuous turnover of OSN and PGCs, the apical dendrites of M/T cells remain stable throughout adulthood over long periods of time (*Mizrahi and Katz, 2003*). Similar dendrite stability has been observed in the projection neurons of the *Drosophila* olfactory system, even when OSNs were ablated in adult flies (*Berdnik et al., 2006*). Our results showed that the refinement of the M/T apical dendrite proceeds normally even when a large fraction of M/T cells are killed soon after birth. In contrast, ablating a large fraction of the M/T cells in early postnatal or adult animals, after the refinement process had occurred and a single dendrite remained projecting to a single glomerulus, induced the surviving M/T cells to expand their dendrites into multiple nearby glomeruli. These observations suggest that reducing the density of M/T cells soon after birth does not interfere with the completion of refinement of apical dendrites, but instead, it is essential for the maintenance of apical dendrite stability as animals mature. Importantly, the main synaptic partners of M/T cells in the glomerulus, the OSNs and juxtaglomerular cells, had not been directly perturbed by our genetic manipulation. This observation suggests that M/T cells apical dendrite stability depends more on the cell density of the peer M/T cells than on their other glomerular synaptic partners (OSN axons, ETCs, PGCs, and SACs). Interestingly, M/T cell ablation in adults after the initial refinement is complete, only induced half of the remaining M/T cells (~50%) to expand their dendritic tuft toward several glomeruli (*Figure 2*). This observation could indicate that different subtypes of M/T cells may respond differently to these perturbations, and is consistent with previous works demonstrating that M/T cells are not a homogeneous cell type (*Angelo et al., 2012*; *Kollo et al., 2014*; *Padmanabhan and Urban, 2010*). Alternatively, the fact that only half of M/T cells undergo structural changes could be merely a stochastic event.

## Mechanisms controlling structural plasticity of the M/T cells in adult mice

A previous work has shown that reducing the number of bipolar cells in mouse embryonic retina induces dendrite expansion in remaining cells when the retinal circuit is being assembled (*Johnson*

*et al., 2017*). The results from our early postnatal cell ablation experiment in the OB suggest that the apical dendrite maturation of M/T cells can proceed normally even when a large fraction of M/T cells are being progressively killed. However, our experiments indicate that cell density plays an important role in maintaining the stability of the apical dendrite of M/T cells. There is evidence that neurotransmission plays a role in the growth and maturation of dendrites and synapses (*Andreae and Burrone, 2015*; *Bleckert and Wong, 2011*; *Rajan et al., 1999*). These previous works would be consistent with a possible scenario where after ablating a large fraction of M/T cells, OSN axons and ETCs cells that do not have a M/T synaptic partner in the glomerulus increased the neurotransmitter levels in the glomerulus, thus triggering the dendritic growth from nearby M/T cells, as we observed in our experiments.

Neuronal activity plays an essential role in sculpting the neuronal circuit as it is being assembled during development (*Andreae and Burrone, 2015*; *Bleckert and Wong, 2011*; *Martini et al., 2021*). However, the refinement of M/T dendrites in early postnatal animals is independent of sensory stimuli (*Lin et al., 2000*). Our experiments indicated that ablating a high percentage of M/T cells in adult animals (in Tbx21::iDTR mice) caused the expansion of apical dendrites into multiple glomeruli, even when sensory input was blocked by naris occlusion (*Figure 3A and B*). This suggests that sensory input is dispensable for M/T cells apical dendrite pruning and stabilization at perinatal stages and adult stages, respectively. In a recent paper (*Fujimoto et al., 2023*), Kir2.1 was expressed in a large percentage of M/T cells during embryonic development to block their electrical activity. This work showed that these embryonic silent M/T cells failed to prune their arbors to a single dendrite. In our experiments, we expressed Kir2.1 in adult animals, and we observed that the ablation of a large fraction of M/T cells still caused the remaining M/T cells to sprout several new tufts towards multiple glomeruli. In aggregate, these observations indicate that action potentials may be necessary for the normal pruning that occurs during early development (*Fujimoto et al., 2023*), but are not required for the expansion of dendritic trees caused by ablating a large fraction of M/T cells in adult animals (this current study).

## Projection neurons are dispensable to form an olfactory map of sensory axons on the olfactory bulb, but are necessary to maintain it during adulthood

The olfactory map formed by OSN axons on the OB is maintained throughout life despite the turnover of OSNs (*Graziadei and Graziadei, 1979*; *Zou et al., 2004*). The flow of information between OSNs and M/T cells is an essential step for the transmission of odor information because M/T cells are the only connection from the OB to the rest of the brain. Several previous works have investigated how OSN manipulations affect the formation and maintenance of the odor map (*Biju et al., 2008*; *Lin et al., 2000*; *Ma et al., 2014*; *Yu et al., 2004*; *Zheng et al., 2000*).Here, we evaluated whether reducing the density of M/T cells affects odor map formation and/or maintenance. Our results showed that odor map formation proceeds normally in animals in which the M/T cells are progressively killed after birth. These results are consistent with previous works that showed that initial convergence of OSN axons expressing the same odorant receptor into single glomeruli does not require M/T cells, as reported in Tbr1 mutant mice that cannot produce M/T cells (*Bulfone et al., 1998*). In contrast, we observed that odor map maintenance requires a full complement of M/T cells (*Figure 4*). With a reduced density of M/T cells the axons of OSNs initially project to a single glomerulus, but as the animals mature, those axons eventually project into multiple neighboring glomeruli. Moreover, in animals that developed a normal olfactory map with a full set of M/T cells, reducing the density of M/T cells as adults also caused OSN axons to project abnormally into multiple glomeruli. Therefore, the present work shows that while M/T cells are not necessary for the formation of a normal odor map on the OB, they are necessary for the maintenance of this map in adult animals.

The odor map disruption observed after ablation of M/T cells is interesting, because many of the other synaptic partners for OSN axons (ETCs and PGCs) were not directly targeted by our genetic manipulation. One may have anticipated that even without M/T cells dendrites, the neurites from ETCs and PGCs could have provided synaptic targets for the OSN axons, and this could have been sufficient to maintain the odor map on the OB surface. Indeed, there is some evidence that OSN axons make strong synapses contact with ETCs, and some works indicate that M/T cells are not the primary synaptic target of OSN axons (*De Saint Jan, 2022*; *De Saint Jan et al., 2009*; *Gire et al.,*

*2012*). The fact that the drastic reduction in the number of M/T cells caused the OSN axons to miss their normal targets in the OB in adult animals suggests that the OSN-M/T cell (but not the OSN-ETC) connections are necessary for the maintenance of the olfactory map during adulthood. There are three likely scenarios that could account for our observations. (i) The sprouting of the apical dendrite tuft of remaining M/T cells to several glomeruli could directly cause OSN axons to project into multiple glomeruli. (ii) Ablating M/T cells may cause changes in other cells that make synapses in the glomeruli (ETCs, PGCs, sAC, etc…), such that the misrouting of OSN axons that we observed may be a secondary effect caused by the elimination of M/T cells. (iii) Ablating a large fraction of M/T cells causes a reduction in the OB volume, which could cause a failure of OSN axons to converge onto single glomeruli, which in turn may induce the expansion of apical dendrite tuft from remaining M/T cells.

## Robustness of learned but frailty of innate olfactory driven behaviors

The OB is considered to be crucial for olfaction because it is the only connection between the olfactory sensory epithelium and the olfactory cortex (*Vinograd et al., 2017*; *Wilson and Mainen, 2006*). However, several works have shown that some olfactory function can persist despite severe perturbations of the OB. For example, mice in which the topographic projection from OSNs to the OB is genetically perturbed can perform simple odor discrimination tasks (*Fadool et al., 2004*; *Fleischmann et al., 2008*). In addition, some rodents with mild or even severe disruption of the OB architecture by NMDA injections or extensive lesions of the OB can still perform odor discrimination tasks, although a certain fraction of these animals become completely anosmic (*Erskine et al., 2019*; *Lu and Slotnick, 1998*). However, the reproducibility of the surgical or chemical lesions makes it difficult to interpret these results. In addition, in bulbectomized rats, the axons from OSN neurons project into rostral olfactory areas, and that these ectopic projections could explain why some of these animals missing an OB can discriminate between odors to some extent (*Slotnick et al., 2004*). Finally, a recent work has reported that some humans in which their OBs cannot be detected by MRI can still smell (*Weiss et al., 2020*).

Our results showed that a drastic reduction of M/T cells density (fewer than 5%) cause a severe disruption of the adult OB architecture: OSNs expressing the same odor receptor fail to converge into a single glomerulus, and the apical dendrites from individual M/T cells innervate multiple glomeruli. However, ablation of M/T cells in adult mice (in Tbx21::iDTR mice) showed that 25% of mice became permanently anosmic, unable to perform odorant detection tests, regardless of how many days they were exposed to the test. In contrast, the remaining ~75% mice retained significant olfactory function, although these animals needed training (*Figure 5*). Remarkably, all adult mice in which M/T cells were ablated soon after birth (Tbx21::DTA) performed well on olfactory tests, even when only 1% of their M/T neurons remained, the only connection between the OB and the olfactory cortex. Moreover, in these mice, the organization of the projections from the OSNs to the glomeruli (the 'odor map'), was severely disrupted, and the dendrites of M/T cells projected abnormally into multiple glomeruli. Overall, these results suggest that the brain of young mice exhibit high resilience in the performance of learned olfactory behaviors, whereas this functional resilience may decrease with age.

Our manipulations cause two major changes in the olfactory system, one primary, and several secondary. The primary change is a large reduction in the number of M/T cells both in the MOB and AOB. This reduction in M/T cell number triggers significant secondary changes in the connectivity of the bulb, including an abnormal projection of OSNs onto the OB, and the growth of ectopic dendrites from the remaining M/T cells into multiple glomeruli. The behavioral abnormalities displayed by these mice is ultimately caused by the reduction in the number of M/T cells, but it is likely that the secondary structural changes could regulate some of the behavioral phenomena that we observed. For example, in principle, it is possible that the ectopic dendrites innervating several glomeruli could help the bulb to perceive smells with a much reduced number of M/T cells. On the other hand, the promiscuous growth of dendrites into multiple glomeruli could make it more difficult for the animals to discriminate between smells. The same argument could be made about the fact that OSN axons project onto multiple glomeruli: we do not know if this change helps or makes it more difficult for the animal to detect smells.

Mice with reduced numbers of M/T cells were able to perform satisfactorily in tests that measured learned behaviors (such as odor detection). However, the performance of these mice with

innate-olfactory-driven behaviors such as mating with females or aggression against intruders was severely impaired. (*Figure 5*). It is generally assumed that learned olfactory behaviors and innate olfactory driven behaviors are primarily controlled by the main and accessory bulbs (MOB and AOB), respectively. However, we observed that our genetic manipulations caused a comparable elimination of M/T cells in the MOB and AOB. Several reasons could explain the different susceptibility of learned or innate olfactory behaviors to ablation of M/T cell. Learned behaviors may be more resilient to experimental perturbations than innate-olfactory-driven behaviors, perhaps indicating that the performance of innate behaviors depends on hardwired circuits that cannot be easily reconfigured. Indeed, this hypothesis is consistent with our observation that ablating M/T cells in adult animals (in Tbx21::iDTR mice) caused an initial failure to perform odor detection tests (a learned behavior), which could be overcome after several days of training. Alternatively, the standard tests used to probe learned olfactory behaviors, such as threshold detection with olfactometers, could be much simpler for animals to perform than the olfactory innate driven behaviors, such as mating or aggression. It is important to note that the procedures commonly used to measure learned olfactory behaviors require substantial training, and thus, are quite artificial. In this scenario, animals with a severe perturbation of the OB may still capable of performing normally on olfactometer tests, but will be unable to perform natural behaviors that require complex olfactory processing.

In summary, our results indicate that the full complement of M/T cells, the projection neurons that transmit odor information from the OB into the rest of the brain, is not required for normal assembly of the olfactory circuit, but it is essential for its maintenance. The initial steps of several processes in the assembly of the OB proceed normally despite a drastically (>95%) reduced number of M/T cells. Early postnatal M/T cell ablation did not affect the appropriate targeting of OSN axons into the OB, or the refinement to a single apical dendrite of remaining M/T cells that projected into a single glomerulus. However, after the OB circuitry was normally assembled despite a reduced density of M/T cells, we observed that the maintenance of the OB wiring was perturbed in multiple ways, including expansion of several dendrites from M/T cells that projected into multiple glomeruli, and misrouting of OSN axons. Importantly, we observed that both of these changes in circuit wiring (apical dendrite expansion and OSN axon misrouting) could also be triggered in animals in which the OB assembly had occurred under normal circumstances, but in which the M/T cell ablation was induced in full adult animals. These experiments highlight that in some brain regions, a full density of cells may be dispensable for the assembly of its neuronal circuits, but it may be required for its maintenance during adulthood. Finally, these observations indicate that olfactory circuits have a remarkable ability to reorganize such that after a severe (>95%) loss of an essential neuronal type during development and adult ages, olfactory function is surprisingly preserved.

## Methods
### Animals
, Rosa26, Rosa26, Or8a1b (M72), Rosa26[Ai9] and Rosa26[Confetti] were obtained from the Jackson laboratory. The Tbx21[Cre] driver mouse (JAX#024507) was used to express the Cre recombinase in M/T cells, the olfactory bulb projection neurons (*Mitsui et al., 2011*). The reporter mouse DTA (JAX#009669) expresses the subunit A of the diphtheria toxin upon cre recombination (*Voehringer et al., 2008*), inducing death of M/T cells at early postnatal stages when crossed with Tbx21[Cre] mouse. The Rosa26[iDTR] reporter mouse (JAX#007900) expresses the diphtheria toxin receptor upon cre recombination (*Buch et al., 2005*), allowing for the controlling the timing of M/T cell ablation after injecting the diphtheria toxin into the Tbx21::iDTR mouse. The Rosa26[Ai9] (JAX#007909) and Rosa26[Confetti] (JAX#017492) reporter mouse expresses the tdTomato fluorescent protein or produces four different fluorescent proteins (RFP, YFP, GFP or CFP) upon cre recombination (*Madisen et al., 2010*; *Snippert et al., 2010*), labeling M/T cells when crossed with the Tbx21[Cre] mouse. The Or8a1b[ChR2-YFP] mouse reporter mouse (JAX#021206) expresses the yellow fluorescent protein in the M72 olfactory sensory neurons (*Smear et al., 2013*), allowing the visualization of the glomeruli innervated by their axons in the OB.

All mice were handled according to the protocol #1709 approved by the California Institute of Technology (Caltech) Institutional Animal Care and Use Committee (IACUC). Mice colonies were maintained at the animal facility at Caltech.

## Ablation of M/T cells

Ablation of early postnatal M/T cells was directly achieved in the transgenic mice Tbx21::DTA, as a result of crossing the *Tbx21*^Cre driver mouse with the reporter *Rosa26*^DTA mouse. Ablation of adult M/T cells was induced by intraperitoneal injection of the diphtheria toxin A (DT) subunit (Sigma D0564) at a concentration of 20 μg/kg body weight into Tbx21::iDTR transgenic mice (***Dong et al., 2021***; ***Tien et al., 2016***), with two injections separated by 24 hr starting at either P60 or P180. To ablate 75% of M/T cells, we used half of the DT dose (10 μg/kg body weight).

## Tissue processing, immunohistochemistry, and imaging

Mice were deeply anesthetized with a single intraperitoneal injection of Ketamine/Xylazine (100 mg/10 mg per animal kg) and fixed by intracardiac perfusion with 4% PFA in 0.1 M phosphate buffer (PBS), pH 7.4. Brains were postfixed overnight in 4% PFA at 4 °C. The next day, all samples were washed three times, 10 min each, with 0.1 M PBS. Then, brains were embedded into 3% agarose, and the OBs were cut in a vibratome into 50-μm-thick sections for immunostaining and cell quantification, and analysis of OSN axon projection into the OB, or into 300-μm-thick sections for M/T cells apical dendrite analyses. Immunohistochemistry was performed in 50-μm-thick sections using the following blocking solution: 10% fetal bovine serum (FBS) and 0.1% Triton X-100 in 0.1 M PBS. Sections were incubated overnight at 4 °C with primary antibodies diluted in blocking solution: rat anti-Tbr2 (1:500, Invitrogen 14-4875-80; RRID:AB_11043546), chicken anti-tyrosine hydroxylase (TH; 1:250, Abcam AB6211; RRID:AB_2240393), mouse anti-calbindin (1:1.000, Synaptic Systems 214 011; RRID:AB_2068201) and rabbit anti-GFP (1:1.000, AB3080P Millipore; RRID:AB_2630379). The next day, sections were washed and incubated with secondary antibody diluted in blocking solution: Goat anti-Rabbit IgG Alexa-488 (Molecular Probes A-11008; RRID:AB_143165), Goat anti-Rat IgG Alexa 594 (Molecular Probes A-11007; RRID:AB_141374), Goat anti-Mouse IgG Alexa 647 (Molecular Probes A-21235; RRID:AB_2535804). 300 μm thick OB sections were clarified using the ScaleSQ(5) method (***Hama et al., 2015***). All sections were incubated with DAPI (D9542, Sigma) to label cell nuclei and treated with TrueBlack (23007, Biotum) to eliminate lipofuscin autofluorescence.

Z-stack images were acquired using 20 x, 40 x, or 60 x objectives on a confocal microscope (Zeiss LSM 800). Z-stacks were merged and analyzed using ImageJ or Imaris and edited with Photoshop (Adobe) software.

## Measurement of OB volume

OBs from Tbx21::Ai9, Tbx21::DTA::Ai9, and Tbx21::iDTR::Ai9 mice were sectioned in 50-μm-thick transversal sections, collected in sequence, and stained with DAPI. The area of each of the OB layers (glomerular, external plexiform, granular, and rostral migratory stream) was calculated using the Neurolucida software (MBF Bioscience Inc, Williston, VT). The volume of each layer was calculated multiplying the area by the thickness of the section (area x 50 μm). Then, the OB volume was the result of the sum of the volumes of each layer of an OB (***Richard et al., 2010***).

## Mounting and analyzing morphology of M/T cells

Quantification of M/T cells was performed on Tbx21::Ai9, Tbx21::DTA::Ai9, and Tbx21::iDTR::Ai9 mice using Neurolucida software. OBs were sectioned into 50-μm-thick transversal sections, collected in sequence and stained with antibodies against RFP to amplify the tdTomato signal. M/T cells were counted in all OB sections from experimental mice (Tbx21::DTA::Ai9 and Tbx21::iDTR::Ai9), while in Tbx21::Ai9 mice, we quantified the number of M/T cells in five alternate sections for each OB (n=3) and extrapolated to the total number of M/T cells in the whole OB from cell density obtained in the analyzed sections (***Richard et al., 2010***).

For analyses of apical dendrites, M/T cells were labeled by systemic intravenous injection of AAV-PHP.eB vectors carrying EGFP gene under the human synapsin promoter. Intravenous administration of AAV-PHP.eB vectors was performed by injection into the retro-orbital sinus of adult mice 2–3 weeks before perfusing animals for histological analysis. In wild-type animals, we performed systemic injections with a two-component AAV-PHP.eB viral vector system to achieve sparse labeling of M/T cells (***Chan et al., 2017***).

### Analysis of M/T cell apical dendrites and OSN axon projections

Confocal images of the M/T cells apical dendrite tuft, taken with a 60 x objective, were analyzed with the filament tracer tool from the Imaris software (https://imaris.oxinst.com/). Mitral and tufted cells were included in the same group without distinguishing between the two cell types. The number of glomeruli innervated by M/T cells apical dendrite or OSN axons was estimated using a confocal microscope and complemented by inspection with a stereo fluorescence microscope.

### Olfactory sensory deprivation and reduction of neuronal activity

#### Naris occlusion

Mice were anesthetized with a single intraperitoneal injection with ketamine/xylazine (100 mg/10 mg per animal kg). Unilateral naris occlusion was performed by cautery with a high-temperature cautery tip (Bovie Aaron Change-A-TipTM) ten days before inducing ablation of M/T cells in Tbx21::iDTR mice by injecting DT intraperitoneally (*Lin et al., 2000*). To confirm the efficiency of naris occlusion, we measured the levels of tyrosine hydroxylase in PGs by immunostaining (*Baker et al., 1993*).

#### Reduction of neuronal activity

AAV-PHP.eB carrying the Kir2.1 gene connected by a 2 A linker to EGFP in a DiO conformation (inverted and flanked by loxP sites) was injected systemically into Tbx21::iDTR mice 15 days before inducing M/T ablation by DT injection. The efficiency of silencing by the Kir2.1 channel in the M/T cells was evaluated by electrophysiological recordings.

#### Slice electrophysiology

Mice were deeply anesthetized by intraperitoneal injection of ketamine/xylazine and perfused transcardially with ice-cold sucrose slicing solution (in mM, sucrose 213, KCl 2.5, $NaH_2PO_4$ 1.2, $NaHCO_3$ 25, glucose 10, $MgSO_4$ 7, $CaCl_2$ 1, pH 7.35). After decapitation, the brain was removed and immersed in the same ice-cold slicing solution. Horizontal slices (300 μm) of OB were cut using a vibratome (VT-1200s, Leica). We first let the slices recover in artificial cerebrospinal fluid (ACSF. NaCl 124, KCl 2.5, $NaH_2PO_4$ 1.2, $NaHCO_3$ 24, glucose 25, $MgSO_4$ 1, CaCl2 2) at 33 °C for 30 min and then held them at room temperature (~22 °C) until use.

During recording, slices were perfused continuously (~2 mL/min) with ACSF at 25 °C. Neurons were visualized and targeted using an upright IR-DIC microscope (BX51WI, Olympus). Mitral cells were identified by their characteristic large soma within the mitral cell layer. Whole-cell recordings were achieved using glass pipettes with an impedance of 3–6 MOhm when filled with intracellular solution (K-gluconate 145, NaCl 2, KCl 4, HEPES 10, EGTA 0.2, Mg-ATP 4, Na-GTP 0.3, pH 7.25). Electrical signals were sampled at 20 kHz and filtered at 2.9 KHz using an EPC-10 amplifier (HEKA Elektronik). Liquid junction potential was not corrected. Passive and active membrane properties were tested by injecting 500 ms-long current steps with amplitudes ranging from 50 to 1000 pA.

### Behavior experiments

#### Learned behavior

The ability of mice to detect odorants was evaluated in a four channel olfactometer (*Slotnick and Bodyak, 2002*), two months after cell ablation. Several days before starting with the go/no-go paradigm training, mice were partially water-deprived (85% of their initial body weight). First, mice were trained to introduce their snout into the odor sampling port and lick the water valve to get the water reward. During this initial phase of training, only mineral oil was presented. Next, we began with the go/no-go paradigm using Cineole (843788, Sigma) as a reinforced stimulus (S+) and mineral oil as an unreinforced stimulus (S−). Every day, mice underwent 10 training blocks (20 trials per block, with 10 trials with a positive stimulus, and 10 trials with a negative stimulus) per day, for a total of 200 trials per day. A single random stimulus was presented as S+or S- in each trial. The accuracy was calculated for each block of 20 trials. Mouse accuracy >80% was considered as successful learning of the positive (S+) and negative (S-) stimuli. During 2 days (40 trials), mice were trained with 5% cineole concentration to ensure that mice can learn the task. After this initial phase, we decreased the concentration of the reward stimulus at 1%, 0.5%, 0.1%, 0.05%, 0.01%, 0.005%, 0.001%, to determine the odor detection threshold.

Odor discrimination was first tested using two different odors: cineole (S+) and eugenol (S-) (818455, Sigma) both diluted at a concentration of 1%. First, pure odors were presented to the mice: only cineole as S+and only eugenol as S-. On subsequent days, a mixture of cineole and eugenol was presented at either 80/20 (80% cineole/20% eugenol for S+, and 80% eugenol/20% cineole for S−), or 55/45 ratios. Finally, we tested a more challenging odor discrimination task using enantiomer molecules: (+)carvone as S+ (, Sigma) and (−)carvone as S− (22060, Sigma) both diluted to 1%.

## Social behavior

For *mating experiments*, male mice without sexual experience were used to test the mice's mating ability when a wild type female was introduced into the male cage and left there for 3 d. Females were exposed to male bedding for 4 d before being introduced to the male cage. This procedure was repeated with two more females (a total of three females). Successful mating was annotated when a vaginal plug was detected in the morning with any of the females presented.

For *aggression experiment*s, male mice were housed with a female for at least 2 wk before the testing day. The resident/intruder paradigm was used to evaluate the male aggressive behavior. A male was introduced to the home cage of the resident male for 10 min. Aggression behavior was considered to occur when the resident male incited the aggression.

## Data analyses

Excel and Matlab were used for statistical analyses. Wilcoxon rank-sum test was used to compare the tuft length of M/T cells apical dendrites (for two groups). $\chi$2 test with Bonferroni correction was used for comparison of M/T cell apical dendrites and OSN axons glomeruli innervation (for more than two groups).

## Acknowledgements

We are grateful to Walter G Gonzalez, Tarciso Velho and Ting Hao Huang for discussion of the experiments.

## Additional information

### Funding

| Funder | Grant reference number | Author |
| --- | --- | --- |
| NIH Office of the Director | R01MH116508A | Carlos Lois |

The funders had no role in study design, data collection and interpretation, or the decision to submit the work for publication.

### Author contributions

Luis Sánchez-Guardado, Conceptualization, Data curation, Formal analysis, Supervision, Validation, Investigation, Visualization, Methodology, Writing – original draft, Writing – review and editing; Peyman Razavi, Investigation, Methodology; Bo Wang, Antuca Callejas-Marín, Formal analysis, Investigation, Methodology; Carlos Lois, Conceptualization, Resources, Formal analysis, Supervision, Funding acquisition, Validation, Investigation, Visualization, Methodology, Writing – original draft, Project administration, Writing – review and editing

### Author ORCIDs

Luis Sánchez-Guardado ⓘ https://orcid.org/0000-0001-5598-8608
Peyman Razavi ⓘ https://orcid.org/0000-0002-2650-152X
Carlos Lois ⓘ https://orcid.org/0000-0002-7305-2317

### Ethics

This study was performed in strict accordance with the recommendations in the Guide for the Care and Use of Laboratory Animals of the National Institutes of Health. All of the animals were handled according to protocols (#1709) approved by the Institutional Animal Care and Use Committee (IACUC)

of the California Institute of Technology (Caltech). Mice colonies were maintained at the Caltech animal facility.

Reviewer #1 (Public review): https://doi.org/10.7554/eLife.90296.3.sa1
Reviewer #2 (Public review): https://doi.org/10.7554/eLife.90296.3.sa2
Author response https://doi.org/10.7554/eLife.90296.3.sa3

## Additional files

### Supplementary files
• MDAR checklist

### Data availability
All data generated or analyzed during this study are included in the manuscript and supporting files.

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
