## [Editor Report · eLife Assessment]

This **valuable** study shows that eliminating a large portion of the principal neurons in the mammalian olfactory bulb does not affect the initial establishment of the circuit but has an impact on its maintenance. The strength of the paper is that the anatomical changes induced by genetic ablation of neurons are clear-cut. There is **solid** support for the findings, with a description of the structural and behavioral effects of ablating the majority of M/T neurons.

---

## [Referee Report · Reviewer #1 (Public review)]

This paper aims to address the establishment and maintenance of neural circuitry in the case of a massive loss of neurons. The authors used genetic manipulations to ablate the principal projection neurons, the mitral/tufted cells, in the mouse olfactory bulb. Using diphtheria toxin (Tbx21-Cre:: loxP-DTA line) the authors ablated progressively large numbers of M/T cells postnatally. By injecting diphtheria toxin (DT) into the Tbx21-Cre:: loxP-iDTR line, the authors were able to control the timing of the ablation in the adult stage. Both methods led to the successful elimination of a majority of M/TCs by 4 months of age. The authors made a few interesting observations. First, they found that the initial pruning of the remaining M/T cell primary dendrite was unaffected. However, in adulthood, a significant portion of these cells extended primary dendrites to innervate multiple glomeruli. Moreover, the incoming olfactory sensory neuron (OSN) axons, as examined for those expressing the M72 receptor, showed a divergent innervation pattern as well. The authors conclude that M/T cell density is required to maintain the dendritic structures and the olfactory map. To address the functional consequences of eliminating a large portion of principal neurons, the authors conducted a series of behavioral assays. They found that learned odor discrimination was largely intact. On the other hand, mating and aggression were reduced. The authors concluded that learned behaviors are more resilient than innate ones.

The study is technically sound, and the results are clear-cut. The most striking result is the contrast between the normal dendritic pruning during early development and the expanded dendritic innervation in adulthood. It is a novel discovery that can lead to further investigation of how the single-glomerulus dendritic innervation is maintained. The authors conducted a few experiments to address potential mechanisms, but it is inconclusive, as detailed below. It is also interesting to see that the massive neuronal loss did not severely impact learned odor discrimination. This result, together with previous studies showing nearly normal odor discrimination in the absence of large portions of the olfactory bulb or scrambled innervation patterns, attests to the redundancy and robustness of the sensory system.

---

## [Referee Report · Reviewer #2 (Public review)]

The authors make the interesting observation that the developmental refinement of apical M/T cell dendrites into individual glomeruli proceeds normally even when the majority of neighboring M/T cells are ablated. At later stages, the remaining neurons develop additional dendrites that invade multiple glomeruli ectopically and, similarly, OSN inputs to glomeruli lose projection specificity as well. The authors conclude that the normal density of M/T neurons is not required for developmental refinement, but rather for maintaining specific connectivity in adults.

Comments on revised submission:

The authors have adjusted the interpretation of their findings and as a consequence, the conclusions are now better supported by the data. However, the evidence for the absence of a role of firing in regulating ectopic dendrites is still insufficient.

---

## [Author Response]

The following is the authors’ response to the original reviews.

**Reviewer #1 (Public Review):**
This paper aims to address the establishment and maintenance of neural circuitry in the case of a massive loss of neurons. The authors used genetic manipulations to ablate the principal projection neurons, the mitral/tufted cells, in the mouse olfactory bulb. Using diphtheria toxin (Tbx21-Cre:: loxP-DTA line) the authors ablated progressively large numbers of M/T cells postnatally. By injecting diphtheria toxin (DT) into the Tbx21-Cre:: loxP-iDTR line, the authors were able to control the timing of the ablation in the adult stage. Both methods led to the successful elimination of a majority of M/TCs by 4 months of age. The authors made a few interesting observations. First, they found that the initial pruning of the remaining M/T cell primary dendrite was unaffected. However, in adulthood, a significant portion of these cells extended primary dendrites to innervate multiple glomeruli. Moreover, the incoming olfactory sensory neuron (OSN) axons, as examined for those expressing the M72 receptor, showed a divergent innervation pattern as well. The authors conclude that M/T cell density is required to maintain the dendritic structures and the olfactory map. To address the functional consequences of eliminating a large portion of principal neurons, the authors conducted a series of behavioral assays. They found that learned odor discrimination was largely intact. On the other hand, mating and aggression were reduced. The authors concluded that learned behaviors are more resilient than innate ones.The study is technically sound, and the results are clear-cut. The most striking result is the contrast between the normal dendritic pruning during early development and the expanded dendritic innervation in adulthood. It is a novel discovery that can lead to further investigation of how the single-glomerulus dendritic innervation is maintained. The authors conducted afew experiments to address potential mechanisms, but it is inconclusive, as detailed below. It is also interesting to see that the massive neuronal loss did not severely impact learned odor discrimination. This result, together with previous studies showing nearly normal odor discrimination in the absence of large portions of the olfactory bulb or scrambled innervation patterns, attests to the redundancy and robustness of the sensory system. The discussion should take into account these other studies in a historical context.Main comments:(1) In previous studies, it has been concluded that dendritic pruning unfolds independently, regardless of the innervation pattern or activity of the OSNs. The new observation bolsters this conclusion by showing that a loss of neighboring M/T cells does not affect the developmental process. A more nuanced discussion comparing the results of these studies would strengthen the paper.

We thank the reviewer for the suggestion. We now include an extended discussion citing relevant previous works in the manuscript (Lines 351-374).

(2) The authors propose that a certain density of M/T is required to prevent the divergent innervation of primary dendrites, but the evidence is not sufficient to support this proposal. The experiment with low-dose DT injection to ablate a smaller portion of M/T cells did not change the percentage of cells innervating two or more glomeruli. The authors suggest that a threshold must be met, but this threshold is not determined.

In our experiments using high-dose DT, we hypothesized that there may be many empty glomeruli (glomeruli not innervated by M/T cells), and as a result, that some of the remaining M/T cells could branch their apical dendrite tuft into multiple empty glomeruli. To test this hypothesis, we carried out another experiment using a lower dose of DT. In this experiment, the fraction of remaining M/T cells was 25% (~10,000 M/T cells), which was higher than with the high DT dose (5%, or around 2,000 M/T cells) , but still significantly lower than wild type mice (~40,000 cells M/T cells). With around 2,000 glomeruli and 10,000 M/T per bulb, it could be expected that each glomerulus would be innervated by ~5 M/T cells (on average). However, we found that the percentage of M/T cells projecting to multiple glomeruli (around 40%) was similar when either 10,000 or 2,000 of M/T remained in the bulb. In addition, it is important to emphasize that even in wt animals with a full set of M/T cells, a small percentage of M/T cells still innervate more than one glomerulus (Lin et al., 2000). Together, these observations suggest that the innervation of multiple glomeruli by M/T cells is not simply due to the presence of empty glomeruli, and that our hypothesis was not correct.

We have added a comment explaining this issue in the Results section (Lines 200-203).

(3) The authors suggest that neural activity is not required for this plasticity. The evidence was derived primarily from naris occlusion and neuronal silencing using Kir2.1. While the results are consistent with the notion, it is a rather narrow interpretation of how neural activity affects circuit configuration. Perturbation of neural activity also entails an increase in firing. Inducing the activity of the neurons may alter this plasticity. Silencing per se may induce a homeostatic response that expands the neurite innervation pattern to increase synaptic input to compensate for the loss of activity. Thus, further silencing the cells may not reduce multiglomerular innervation, but an increased activity may.

The experiments with Kir2.1 demonstrate that the structural plasticity observed after reducing the total number of M/T cells in an animal is not regulated by the firing action potentials in the remaining cells. Instead, this experiment indicates that the observed structural plasticity may be regulated by other types of mechanisms (including increased synaptic excitation as suggested by the reviewer) that do not require the firing of action potentials in M/T cells.

We now have included a comment regarding this point (Lines 243-247).

(4) There is a discrepancy between this study and the one by Fujimoto et al. (Developmental Cell; 2023), which shows that not only glutamatergic inputs to the primary dendrite can facilitate pruning of remaining dendrites but also Kir2.1 overexpression can significantly perturb dendritic pruning. This discrepancy is not discussed by the authors.

We agree that it would be useful to contrast these two works.

In our experiments, performed in adult animals, we blocked sensory input by performing naris occlusion before we induced ablation of M/T cells. In a separate experiment, also in adult animals, we expressed the Kir2.1 channel, to reduce the ability of neurons to fire action potentials. With both types of manipulations, we observed that the ablation of a large fraction of M/T cells still caused the remaining M/T cells to maintain a single apical dendrite that sprouts several new tufts towards multiple glomeruli. A recent paper (Fujimoto et al., 2023) in which Kir2.1 was expressed in a large percentage of M/T starting during embryonic development showed that these “silent” M/T cells failed to prune their arbors to a single dendrite. In aggregate, these observations indicate that action potentials are necessary for the normal pruning that occurs during perinatal development (Fujimoto et al., 2023), but are not required for the expansion of dendritic trees caused by ablating a large fraction of M/T cells in adult animals (our current manuscript).

We have now explained the differences between both studies in the manuscript (Lines 427-439).

(5) An alternative interpretation of the discrepancy between the apparent normal pruning by p10 and expanded dendritic innervation in adulthood is that there are more cells before P10, when ~25% of M/T cells are present, but at a later date only 1-3% are present.The relationship between the number of M/T cells and single glomerulus innervation has not been explored during postnatal development. It would be important to test this hypothesis.

We agree with this comment, and in lines 375-381 we discuss the discrepancy between normal refinement during development, and dendritic sprouting in adults.

Cre is expressed in M/T cells and it induces DTA expression starting around P0. The elimination of M/T cells starts at this time, and continues until by P10, when more than 75% of M/T have been eliminated. At P21 more than 90% of M/T have been eliminated, and their number remains stable thereafter.

Pruning of the dendrites of M/T cells starts at P0 and it is mostly complete by P10. Therefore, it is possible that between P0 and P7, when dendrites are being pruned, the number of M/T cells remaining in the bulb is still over a threshold that does not interfere with the process of normal dendrite pruning. We agree that it would be very informative to perform additional experiments in the future where a large set of M/T cells could be ablated before pruning occurs (ideally before P0).

(6) The authors attribute the change in the olfactory map to the loss of M/T cells. Another obvious possibility is that the diffused projection is a response to the change in the olfactory bulb size. With less space to occupy, the axons may be forced to innervate neighboring glomeruli. It is not known how the total number of glomeruli is affected. This question could be addressed by tracking developmental changes in bulb volume and glomerular numbers.

Certainly, this is a possibility, and we have now included a comment on this regard in the manuscript (Lines 473-480).

We believe that there are three likely scenarios that could account for these observations:

(a) After ablating M/T cells, the tufts of the remaining M/T cells sprout into multiple glomeruli, and this causes the axons of OSNs to project into multiple glomeruli.

(b) Ablating M/T cells may cause changes in other OB cells that make synapses in the glomeruli (ETCs, PGCs, sAC, etc…), and the misrouting of OSN axons that we observed in our experiments may be a secondary effect caused by the elimination of M/T cells.

(c) After ablating the majority of M/T cells, the olfactory bulb gets reduced in size, and the axons of OSNs find it difficult to precisely converge on a target that now has become smaller. As a result, the axons of OSNs fail to converge on single glomeruli.

(7) The retained ability to discriminate odors upon reinforced training is not surprising in light of a number of earlier studies. For example, Slotnick and colleagues have shown that rats losing ~90% of the OB can retain odor discrimination. Weiss et al have shown that humans without an olfactory bulb can perform normal olfactory tasks. Gronowitz et al have used theoretical prediction and experimental results to demonstrate that perturbing the olfactory map does not have a major impact on olfactory discrimination. Fleischmann et al have shown that mice with a monoclonal nose can discriminate odors. The authors should discuss their results in these contexts.

We apologize for this important oversight - we now include a more elaborate discussion including the relevant references as suggested in the manuscript (Line 483-496).

(8) It should be noted that odor discrimination resulting from reinforcement training does not mean normal olfactory function. It is a highly artificial situation as the animals are overtrained. It should not be used as a measure of the robustness of the olfactory sense. Natural odor discrimination (without training), detection threshold, and innate appetitive/aversive response to certain odors may be affected. These experiments were not conducted.

We agree that the standard tests commonly used to measure olfactory function require substantial training, and thus, are quite artificial. However, these tests are used because they allow a more precise quantification of olfactory function than those relying on natural behaviors.

We have now included a few sentences to address this point in the results (Lines 321322) and discussion sections (Lines 541-543).

(9) The social behaviors were conducted using relatively coarse measures (vaginal plug and display of aggression). Moreover, these behaviors are most likely affected by the disruption of the AOB mitral cells and have little to do with the dendritic pruning process described in the paper. It is misleading to lump social behaviors with innate responses to odors.

This point follows the same logic as the previous one. The olfactory tests that rely on natural behaviors are quite coarse and difficult to quantify. In contrast, the olfactory tests using apparatuses such as olfactometers can be quantified with precision, but they are artificial. We agree that some of the naturalistic behaviors that we studied such as mating or aggression may depend to a large extent on the AOB (although it is possible that the MOB may also be involved in these tasks to a degree). In our initial version of the manuscript, we commented on the anticipated relative involvement of the MOB and AOB in the studied tasks, but we have now added some additional sentences to make this point clearer. In addition, we now add a comment indicating that it is possible that the abnormal behaviors could simply be due to a reduction in the number of AOB M/T cells (~98.5% and ~ 85% elimination of M/T cells in the AOB in Tbx::DTA and Tbx::iDTR mice, respectively), regardless of the abnormal dendritic pruning of main OB M/T cells (Lines 530-534).

See Figure 5E - M/T cells in AOB (Lines 1238-1239).

**Reviewer #2 (Public Review):**
The authors make the interesting observation that the developmental refinement of apical M/T cell dendrites into individual glomeruli proceeds normally even when the majority of neighboring M/T cells are ablated. At later stages, the remaining neurons develop additional dendrites that invade multiple glomeruli ectopically, and similarly, OSN inputs to glomeruli lose projection specificity as well. The authors conclude that the normal density of M/T neurons is not required for developmental refinement, but rather for maintaining specific connectivity in adults.The observations are indeed quite striking; however, the authors' conclusions are not entirely supported by the data.(1) It is unclear whether the expression of diphtheria toxin that eventually leads to the ablation of the large majority of M/T neurons compromises the cell biology of the remaining ones.

DT is an extremely potent toxin that kills cells by inhibiting proteins translation, and it has been demonstrated that the presence of a single DT molecule in a cell is sufficient to kill it, because of its highly efficient catalytic activity. Accordingly, previous experiments have shown that DT kills cells within a few hours after its appearance in the cytoplasm (Yamaizumi et al., 1978). In other words, all the published evidence suggests that if a cell is exposed to the action of DT, that cell will die shortly. There is no evidence that cells exposed to DT can survive and experience long-term effects. Finally, previous works have not observed any long-term changes in neurons directly caused by the actions of DT (Johnson et al., 2017).

(2) The authors interpret the growth of ectopic dendrites later in life as a lack of maintenance of dendrite structure; however, maybe the observed changes reflect actually adaptations that optimize wiring for extremely low numbers of M/T neurons. The finding that olfactory behavior was less affected than predicted supports this interpretation.

We do not know the cellular or molecular mechanisms that explain why reducing the density of M/T cells is followed by the growth of ectopic dendrites from the remaining M/T cells. We agree that the functional outcome of growing ectopic dendrites may result in an optimization of wiring in the bulb and could explain why olfactory function is relatively preserved. We now include a comment regarding this possibility (Lines 513-525).

(3) The number of remaining M/T neurons is much higher at P10 than later. Can the relatively large number of remaining neurons (or their better health status) be the reason that dendrites refine normally at the early developmental stages rather than a (currently unknown) developmental capacity that preserves refinement?

We thank the reviewer for the suggestion, which was also raised by reviewer 1.

We agree with this comment, and in lines 375-381 we discuss the discrepancy between normal refinement during development, and dendritic sprouting in adults.

Cre is expressed in M/T cells and it induces DTA expression starting around P0. The elimination of M/T cells starts at this time, and continues until by P10, when more than 75% of M/T have been eliminated. At P21 more than 90% of M/T have been eliminated, and their number remains stable thereafter.

Pruning of the dendrites of M/T cells starts at P0 and it is mostly complete by P10. Therefore, it is possible that between P0 and P7, when dendrites are being pruned, the number of M/T cells remaining in the bulb is still over a threshold that does not interfere with the process of normal dendrite pruning. We agree that it would be very informative to perform additional experiments in the future where a large set of M/T cells could be ablated before pruning occurs (ideally before P0).

(4) While the effect of reduced M/T neuron density on both M/T dendrites and OSN axons is described well, the relationship between both needs to be characterized better: Is one effect preceding the other or do they occur simultaneously? Can one be the consequence of the other?

Previous works have demonstrated that disrupting the topographic projection of the OSN axons has no effect on the structure of the apical dendrite of M/T cells (Ma et al., 2014; Nishizumi et al., 2019). Our experiments ablating a large fraction of M/T cells suggest that they are necessary for the correct targeting of OSN axons into the bulb. However, our experiments do not allow us to tell apart these 2 scenarios:

(a) the ablation of a large fraction of M/T cells directly causes the sprouting of the apical dendrite of M/T cells, and that this sprouting in turn causes the abnormal projection of OSN axons onto the bulb.

(b) the ablation of a large fraction of M/T cells first causes the axons of OSN to project abnormally onto multiple glomeruli in the bulb, and this in turn causes the dendrite of remaining M/T cells to sprout onto multiple glomeruli.

We now include a comment on the manuscript explaining this point. (Lines 473-492)

(5) Page 7: the observation that not all neurons develop additional dendrites is not a sign of differences between cell types, it may be purely stochastic.

This is correct, and we mention these 2 scenarios in the discussion (Line 407-408).

(6) Page 8: the fact that activity blockade did not affect the formation of ectopic dendrites does not suggest that the process is not activity-dependent: both manipulations have the same effect and may just mask each other.

The experiments with Kir2.1 demonstrate that the structural plasticity observed after reducing the total number of M/T cells in an animal is not regulated by the firing action potentials in the remaining cells. Instead, this experiment indicates that the observed structural plasticity may be regulated by other types of mechanisms (including increased synaptic excitation as suggested by the reviewer) that do not require the firing of action potentials in M/T cells.

We now have included a comment regarding this point (Lines 243-247).

(7) It remains unclear how the observed structural changes can explain the behavioral effects.

We agree that the relationship between structural changes and behavior was not appropriately explained in our manuscript. Our manipulations cause two major changes in the olfactory system, one primary, and several secondary. The primary change is a large reduction in the number of M/T cells both in the MOB and AOB. This reduction in M/T cell number triggers significant secondary changes in the connectivity of the bulb, including an abnormal projection of OSNs onto the OB, and the growth of ectopic dendrites from the remaining M/T cells into multiple glomeruli.

The behavioral abnormalities displayed by these mice is ultimately caused by the reduction in the number of M/T cells, but it is likely that the secondary structural changes could regulate some of the behavioral phenomena that we observed. For example, in principle, it is possible that the ectopic dendrites innervating several glomeruli could help the bulb to perceive smells with a much reduced number of M/T cells. On the other hand, this promiscuous growth of dendrites into multiple glomeruli could make it more difficult for the animals to discriminate between smells. The same argument could be made about the fact that OSN axons project onto multiple glomeruli: we simply do not know if this change helps or makes it more difficult for the animal to detect smells.

We now include a comment regarding this issue (Lines 513-525).

**Reviewer #1 (Recommendations For The Authors):**
Additional experiments and a more thorough discussion of the results, as suggested in the public review, would significantly strengthen the paper. Below are some specific parts that need to be addressed.There is a lack of information on how M/T cell numbers are quantified. Without the information, it is difficult to evaluate the claim. Using the tdTomato signal may miss cells that are not labeled due to the transgenic effect.

Although we cannot conclude that we are identifying the complete set of M/T cells (because the transgenic lines may fail to label some M/T cells), the number of M/T cells that we observed is similar to that previously reported (Richard et al., 2010). This concern has been included in the Results section (Lines 121-124).

A more detailed description about M/T cells quantification has been added into the method section (Lines 627-632).

There is a lack of information on the timeline of treatment and how measurement of the olfactory bulb volume is conducted.

We now include a more detailed description of how the volume of the OB was measured in the methods (Lines 621-623).

The volume measurement is inconsistent with the pictures shown. In Figure 1, supplemental data 2 panels B and C, it appears that the bulbs in DTA and DTR mice are about half in length in each dimension. This would translate into ~1/8 of the volume of the control mice.

We measured the volume of the bulbs based on the Neurolucida reconstructions, and we observed that in both DTA and iDTR mice the volumes of their bulbs are roughly 50% compared to a wild type mouse. In Figure 1 - figure supplement 2 the sections that were shown for wild type, DTA and iDTR mice were not taken at the same position in the bulb, and this gave the impression that the bulbs from DTA and iDTR were much smaller than they really are. We now show sections for these three animals at equivalent positions in the bulb.

Figure 1 E and F have no legend.

We apologize for this mistake - we have now added the legend for Figures 1E and F (Lines 1009-1013).

Figure 3, supplemental data 2, it is not clear what the readers should be looking at. The data is confusing even for experts in the field. The authors should describe the figures more clearly, pointing out what they are supposed to show.

We apologize for this, and we have now added a more detailed description of Figure3 – figure supplement 2 (Lines 1153-1167).

In several figures, it is not clearly written what the comparisons were for where there are indications of statistical significance above the bars.

We have now included a more detailed description of the statistics comparison in the figure legends.

AAV serotype should be specified.

The AAV serotype used to label M/T cells was the AAV-PHP.eB. We have added this information in the methods section of the manuscript.

**Reviewer #2 (Recommendations For The Authors):**
Minor pointsPage 5, para 2: "The decrease in neuronal plasticity with age": it is unclear what "the decrease" refers to.

We have changed this sentence in the text to make it clear:

“The decrease in structural plasticity of M/T cells after apical dendrite refinement (Mizrahi and Katz, 2003),….”

Line 146-148

Is there a quantification of the effect of Kir2.1 overexpression alone (example shown in Figure 3D)?

We did an experiment in IDTR animals in which a fraction of M/T cells expressed Kir2.1, and we split these animals in 2 groups: (a) animals that received an injection of DT, and (b) animals that did not receive any DT. We quantified the effect of Kir2.1 on M/T cells from animals that received DT injection (with an ablation of around of 90% of M/T cells) and we did not observe any clear statistically significant differences between cells expressing Kir2.1 or neurons that did not express Kir2.1 from other iDTR animals that also received DT injections. We did not quantify the possible effects of kir2.1 in the group of animals that did not receive DT because on a first inspection we did not observe any clear differences between Kir2.1 cells and neighboring wild type cells.

References

Fujimoto S, Leiwe MN, Aihara S, Sakaguchi R, Muroyama Y, Kobayakawa R, Kobayakawa K, Saito T, Imai T. 2023. Activity-dependent local protection and lateral inhibition control synaptic competition in developing mitral cells in mice. Dev Cell S1534-5807(23)00237-X. doi:10.1016/j.devcel.2023.05.004

Johnson RE, Tien N-W, Shen N, Pearson JT, Soto F, Kerschensteiner D. 2017. Homeostatic plasticity shapes the visual system’s first synapse. Nat Commun 8:1220. doi:10.1038/s41467-017-01332-7

Lin DM, Wang F, Lowe G, Gold GH, Axel R, Ngai J, Brunet L. 2000. Formation of precise connections in the olfactory bulb occurs in the absence of odorant-evoked neuronal activity. Neuron **26**:69–80. doi:10.1016/s0896-6273(00)81139-3

Ma L, Wu Y, Qiu Q, Scheerer H, Moran A, Yu CR. 2014. A developmental switch of axon targeting in the continuously regenerating mouse olfactory system. Science 344:194–197. doi:10.1126/science.1248805

Nishizumi H, Miyashita A, Inoue N, Inokuchi K, Aoki M, Sakano H. 2019. Primary dendrites of mitral cells synapse unto neighboring glomeruli independent of their odorant receptor identity. *Commun Biol*
**2**:1–12. doi:10.1038/s42003-018-0252-y

Richard MB, Taylor SR, Greer CA. 2010. Age-induced disruption of selective olfactory bulb synaptic circuits. *Proc Natl Acad Sci U S A*
**107**:15613–15618. doi:10.1073/pnas.1007931107

Yamaizumi M, Mekada E, Uchida T, Okada Y. 1978. One molecule of diphtheria toxin fragment A introduced into a cell can kill the cell. *Cell*
**15**:245–250. doi:10.1016/0092-8674(78)90099-5